# Natural-Parameter Networks:
# A Class of Probabilistic Neural Networks

**Hao Wang, Xingjian Shi, Dit-Yan Yeung**
Hong Kong University of Science and Technology
{hwangaz,xshiab,dyyeung}@cse.ust.hk

## Abstract

Neural networks (NN) have achieved state-of-the-art performance in various applications. Unfortunately in applications where training data is insufficient, they are often prone to overfitting. One effective way to alleviate this problem is to exploit the Bayesian approach by using Bayesian neural networks (BNN). Another shortcoming of NN is the lack of flexibility to customize different distributions for the weights and neurons according to the data, as is often done in probabilistic graphical models. To address these problems, we propose a class of probabilistic neural networks, dubbed natural-parameter networks (NPN), as a novel and lightweight Bayesian treatment of NN. NPN allows the usage of arbitrary exponential-family distributions to model the weights and neurons. Different from traditional NN and BNN, NPN takes distributions as input and goes through layers of transformation before producing distributions to match the target output distributions. As a Bayesian treatment, efficient backpropagation (BP) is performed to learn the natural parameters for the distributions over both the weights and neurons. The output distributions of each layer, as byproducts, may be used as second-order representations for the associated tasks such as link prediction. Experiments on real-world datasets show that NPN can achieve state-of-the-art performance.

## 1   Introduction

Recently neural networks (NN) have achieved state-of-the-art performance in various applications ranging from computer vision [12] to natural language processing [20]. However, NN trained by stochastic gradient descent (SGD) or its variants is known to suffer from overfitting especially when training data is insufficient. Besides overfitting, another problem of NN comes from the underestimated uncertainty, which could lead to poor performance in applications like active learning.

Bayesian neural networks (BNN) offer the promise of tackling these problems in a principled way. Early BNN works include methods based on Laplace approximation [16], variational inference (VI) [11], and Monte Carlo sampling [18], but they have not been widely adopted due to their lack of scalability. Some recent advances in this direction seem to shed light on the practical adoption of BNN. [8] proposed a method based on VI in which a Monte Carlo estimate of a lower bound on the marginal likelihood is used to infer the weights. Recently, [10] used an online version of expectation propagation (EP), called 'probabilistic back propagation' (PBP), for the Bayesian learning of NN, and [4] proposed 'Bayes by Backprop' (BBB), which can be viewed as an extension of [8] based on the 'reparameterization trick' [13]. More recently, an interesting Bayesian treatment called 'Bayesian dark knowledge' (BDK) was designed to approximate a teacher network with a simpler student network based on stochastic gradient Langevin dynamics (SGLD) [1].

Although these recent methods are more practical than earlier ones, several outstanding problems remain to be addressed: (1) most of these methods require sampling either at training time [8, 4, 1] or at test time [4], incurring much higher cost than a 'vanilla' NN; (2) as mentioned in [1], methods

based on online EP or VI do not involve sampling, but they need to compute the predictive density by integrating out the parameters, which is computationally inefficient; (3) these methods assume Gaussian distributions for the weights and neurons, allowing no flexibility to customize different distributions according to the data as is done in probabilistic graphical models (PGM).

To address the problems, we propose *natural-parameter networks* (NPN) as a class of probabilistic neural networks where the input, target output, weights, and neurons can all be modeled by arbitrary exponential-family distributions (e.g., Poisson distributions for word counts) instead of being limited to Gaussian distributions. Input distributions go through layers of linear and nonlinear transformation deterministically before producing distributions to match the target output distributions (previous work [21] shows that providing distributions as input by corrupting the data with noise plays the role of regularization). As byproducts, output distributions of intermediate layers may be used as second-order representations for the associated tasks. Thanks to the properties of the exponential family [3, 19], distributions in NPN are defined by the corresponding natural parameters which can be learned efficiently by backpropagation. Unlike [4, 1], NPN explicitly propagates the estimates of uncertainty back and forth in deep networks. This way the uncertainty estimates for each layer of neurons are readily available for the associated tasks. Our experiments show that such information is helpful when neurons of intermediate layers are used as representations like in autoencoders (AE). In summary, our main contributions are:

- We propose NPN as a class of probabilistic neural networks. Our model combines the merits of NN and PGM in terms of computational efficiency and flexibility to customize the types of distributions for different types of data.
- Leveraging the properties of the exponential family, some sampling-free backpropagation-compatible algorithms are designed to efficiently learn the distributions over weights by learning the natural parameters.
- Unlike most probabilistic NN models, NPN obtains the uncertainty of intermediate-layer neurons as byproducts, which provide valuable information to the learned representations. Experiments on real-world datasets show that NPN can achieve state-of-the-art performance on classification, regression, and unsupervised representation learning tasks.

## 2 Natural-Parameter Networks

The exponential family refers to an important class of distributions with useful algebraic properties. Distributions in the exponential family have the form $p(x|\boldsymbol{\eta}) = h(x)g(\boldsymbol{\eta})\exp\{\boldsymbol{\eta}^T u(x)\}$, where $x$ is the random variable, $\boldsymbol{\eta}$ denotes the natural parameters, $u(x)$ is a vector of sufficient statistics, and $g(\boldsymbol{\eta})$ is the normalizer. For a given type of distributions, different choices of $\boldsymbol{\eta}$ lead to different shapes. For example, a univariate Gaussian distribution with $\boldsymbol{\eta} = (c, d)^T$ corresponds to $\mathcal{N}(-\frac{c}{2d}, -\frac{1}{2d})$.

Motivated by this observation, in NPN, only the natural parameters need to be learned to model the distributions over the weights and neurons. Consider an NPN which takes a vector random distribution (e.g., a multivariate Gaussian distribution) as input, multiplies it by a matrix random distribution, goes through nonlinear transformation, and outputs another distribution. Since all three distributions in the process can be specified by their natural parameters (given the types of distributions), learning and prediction of the network can actually operate in the space of natural parameters. For example, if we use element-wise (factorized) gamma distributions for both the weights and neurons, the NPN counterpart of a vanilla network only needs twice the number of free parameters (weights) and neurons since there are two natural parameters for each univariate gamma distribution.

### 2.1 Notation and Conventions

We use boldface uppercase letters like $\mathbf{W}$ to denote matrices and boldface lowercase letters like $\mathbf{b}$ for vectors. Similarly, a boldface number (e.g., $\mathbf{1}$ or $\mathbf{0}$) represents a row vector or a matrix with identical entries. In NPN, $\mathbf{o}^{(l)}$ is used to denote the values of neurons in layer $l$ before nonlinear transformation and $\mathbf{a}^{(l)}$ is for the values after nonlinear transformation. As mentioned above, NPN tries to learn *distributions* over variables rather than *variables* themselves. Hence we use letters *without* subscripts $c$, $d$, $m$, and $s$ (e.g., $\mathbf{o}^{(l)}$ and $\mathbf{a}^{(l)}$) to denote 'random variables' with corresponding distributions. Subscripts $c$ and $d$ are used to denote natural parameter pairs, such as $\mathbf{W}_c$ and $\mathbf{W}_d$. Similarly, subscripts $m$ and $s$ are for mean-variance pairs. Note that for clarity, many operations used below are implicitly element-wise, for example, the square $\mathbf{z}^2$, division $\frac{\mathbf{z}}{\mathbf{b}}$, partial derivative $\frac{\partial \mathbf{z}}{\partial \mathbf{b}}$, the

gamma function $\Gamma(\mathbf{z})$, logarithm $\log \mathbf{z}$, factorial $\mathbf{z}!$, $1 + \mathbf{z}$, and $\frac{1}{\mathbf{z}}$. For the data $\mathcal{D} = \{(\mathbf{x}_i, \mathbf{y}_i)\}_{i=1}^N$, we set $\mathbf{a}_m^{(0)} = \mathbf{x}_i, \mathbf{a}_s^{(0)} = \mathbf{0}$ (Input distributions with $\mathbf{a}_s^{(0)} \neq \mathbf{0}$ resemble AE's denoising effect.) as input of the network and $\mathbf{y}_i$ denotes the output targets (e.g., labels and word counts). In the following text we drop the subscript $i$ (and sometimes the superscript $(l)$) for clarity. The bracket $(\cdot, \cdot)$ denotes concatenation or pairs of vectors.

## 2.2 Linear Transformation in NPN

Here we first introduce the linear form of a general NPN. For simplicity, we assume distributions with two natural parameters (e.g., gamma distributions, beta distributions, and Gaussian distributions), $\boldsymbol{\eta} = (c, d)^T$, in this section. Specifically, we have factorized distributions on the weight matrices, $p(\mathbf{W}^{(l)}|\mathbf{W}_c^{(l)}, \mathbf{W}_d^{(l)}) = \prod_{i,j} p(\mathbf{W}_{ij}^{(l)}|\mathbf{W}_{c,ij}^{(l)}, \mathbf{W}_{d,ij}^{(l)})$, where the pair $(\mathbf{W}_{c,ij}^{(l)}, \mathbf{W}_{d,ij}^{(l)})$ is the corresponding natural parameters. For $\mathbf{b}^{(l)}$, $\mathbf{o}^{(l)}$, and $\mathbf{a}^{(l)}$ we assume similar factorized distributions.

In a traditional NN, the linear transformation follows $\mathbf{o}^{(l)} = \mathbf{a}^{(l-1)}\mathbf{W}^{(l)} + \mathbf{b}^{(l)}$ where $\mathbf{a}^{(l-1)}$ is the output from the previous layer. In NN $\mathbf{a}^{(l-1)}$, $\mathbf{W}^{(l)}$, and $\mathbf{b}^{(l)}$ are deterministic variables while in NPN they are exponential-family distributions, meaning that the result $\mathbf{o}^{(l)}$ is also a distribution. For convenience of subsequent computation it is desirable to approximate $\mathbf{o}^{(l)}$ using another exponential-family distribution. We can do this by matching the mean and variance. Specifically, after computing $(\mathbf{W}_m^{(l)}, \mathbf{W}_s^{(l)}) = f(\mathbf{W}_c^{(l)}, \mathbf{W}_d^{(l)})$ and $(\mathbf{b}_m^{(l)}, \mathbf{b}_s^{(l)}) = f(\mathbf{b}_c^{(l)}, \mathbf{b}_d^{(l)})$, we can get $\mathbf{o}_c^{(l)}$ and $\mathbf{o}_d^{(l)}$ through the mean $\mathbf{o}_m^{(l)}$ and variance $\mathbf{o}_s^{(l)}$ of $\mathbf{o}^{(l)}$ as follows:

$$(\mathbf{a}_m^{(l-1)}, \mathbf{a}_s^{(l-1)}) = f(\mathbf{a}_c^{(l-1)}, \mathbf{a}_d^{(l-1)}), \quad \mathbf{o}_m^{(l)} = \mathbf{a}_m^{(l-1)}\mathbf{W}_m^{(l)} + \mathbf{b}_m^{(l)}, \tag{1}$$

$$\mathbf{o}_s^{(l)} = \mathbf{a}_s^{(l-1)}\mathbf{W}_s^{(l)} + \mathbf{a}_s^{(l-1)}(\mathbf{W}_m^{(l)} \circ \mathbf{W}_m^{(l)}) + (\mathbf{a}_m^{(l-1)} \circ \mathbf{a}_m^{(l-1)})\mathbf{W}_s^{(l)} + \mathbf{b}_s^{(l)}, \tag{2}$$

$$(\mathbf{o}_c^{(l)}, \mathbf{o}_d^{(l)}) = f^{-1}(\mathbf{o}_m^{(l)}, \mathbf{o}_s^{(l)}), \tag{3}$$

where $\circ$ denotes the element-wise product and the bijective function $f(\cdot, \cdot)$ maps the natural parameters of a distribution into its mean and variance (e.g., $f(c, d) = (\frac{c+1}{-d}, \frac{c+1}{d^2})$ in gamma distributions). Similarly we use $f^{-1}(\cdot, \cdot)$ to denote the inverse transformation. $\mathbf{W}_m^{(l)}$, $\mathbf{W}_s^{(l)}$, $\mathbf{b}_m^{(l)}$, and $\mathbf{b}_s^{(l)}$ are the mean and variance of $\mathbf{W}^{(l)}$ and $\mathbf{b}^{(l)}$ obtained from the natural parameters. The computed $\mathbf{o}_m^{(l)}$ and $\mathbf{o}_s^{(l)}$ can then be used to recover $\mathbf{o}_c^{(l)}$ and $\mathbf{o}_d^{(l)}$, which will subsequently facilitate the feedforward computation of the nonlinear transformation described in Section 2.3.

## 2.3 Nonlinear Transformation in NPN

After we obtain the linearly transformed distribution over $\mathbf{o}^{(l)}$ defined by natural parameters $\mathbf{o}_c^{(l)}$ and $\mathbf{o}_d^{(l)}$, an element-wise nonlinear transformation $v(\cdot)$ (with a well defined inverse function $v^{-1}(\cdot)$) will be imposed. The resulting activation distribution is $p_a(\mathbf{a}^{(l)}) = p_o(v^{-1}(\mathbf{a}^{(l)}))|v^{-1'}(\mathbf{a}^{(l)})|$, where $p_o$ is the factorized distribution over $\mathbf{o}^{(l)}$ defined by $(\mathbf{o}_c^{(l)}, \mathbf{o}_d^{(l)})$.

Though $p_a(\mathbf{a}^{(l)})$ may not be an exponential-family distribution, we can approximate it with one, $p(\mathbf{a}^{(l)}|\mathbf{a}_c^{(l)}, \mathbf{a}_d^{(l)})$, by matching the first two moments. Once the mean $\mathbf{a}_m^{(l)}$ and variance $\mathbf{a}_s^{(l)}$ of $p_a(\mathbf{a}^{(l)})$ are obtained, we can compute corresponding natural parameters with $f^{-1}(\cdot, \cdot)$ (approximation accuracy is sufficient according to preliminary experiments). The feedforward computation is:

$$\mathbf{a}_m = \int p_o(\mathbf{o}|\mathbf{o}_c, \mathbf{o}_d)v(\mathbf{o})d\mathbf{o}, \quad \mathbf{a}_s = \int p_o(\mathbf{o}|\mathbf{o}_c, \mathbf{o}_d)v(\mathbf{o})^2 d\mathbf{o} - \mathbf{a}_m^2, \quad (\mathbf{a}_c, \mathbf{a}_d) = f^{-1}(\mathbf{a}_m, \mathbf{a}_s). \tag{4}$$

Here the key computational challenge is computing the integrals in Equation (4). Closed-form solutions are needed for their efficient computation. If $p_o(\mathbf{o}|\mathbf{o}_c, \mathbf{o}_d)$ is a Gaussian distribution, closed-form solutions exist for common activation functions like $\tanh(x)$ and $\max(0, x)$ (details are in Section 3.2). Unfortunately this is not the case for other distributions. Leveraging the convenient form of the exponential family, we find that it is possible to design activation functions so that the integrals for non-Gaussian distributions can also be expressed in closed form.

**Theorem 1.** *Assume an exponential-family distribution $p_o(x|\boldsymbol{\eta}) = h(x)g(\boldsymbol{\eta})\exp\{\boldsymbol{\eta}^T u(x)\}$, where the vector $u(x) = (u_1(x), u_2(x), \dots, u_M(x))^T$ (M is the number of natural parameters). If activation function $v(x) = r - q\exp(-\tau u_i(x))$ is used, the first two moments of $v(x)$, $\int p_o(x|\boldsymbol{\eta})v(x)dx$*

Table 1: Activation Functions for Exponential-Family Distributions

| Distribution | Probability Density Function | Activation Function | Support |
|---|---|---|---|
| Beta Distribution | $p(x) = \frac{\Gamma(c+d)}{\Gamma(c)\Gamma(d)} x^{c-1}(1-x)^{d-1}$ | $qx^\tau, \tau \in (0,1)$ | $[0,1]$ |
| Rayleigh Distribution | $p(x) = \frac{x}{\sigma^2}\exp\{-\frac{x^2}{2\sigma^2}\}$ | $r - q\exp\{-\tau x^2\}$ | $(0,+\infty)$ |
| Gamma Distribution | $p(x) = \frac{1}{\Gamma(c)} d^c x^{c-1}\exp\{-dx\}$ | $r - q\exp\{-\tau x\}$ | $(0,+\infty)$ |
| Poisson Distribution | $p(x) = \frac{c^x \exp\{-c\}}{x!}$ | $r - q\exp\{-\tau x\}$ | Nonnegative interger |
| Gaussian Distribution | $p(x) = (2\pi\sigma^2)^{-\frac{1}{2}}\exp\{-\frac{1}{2\sigma^2}(x-\mu)^2\}$ | ReLU, tanh, and sigmoid | $(-\infty,+\infty)$ |

and $\int p_o(x|\boldsymbol{\eta})v(x)^2 dx$, can be expressed in closed form. Here $i \in \{1, 2, \ldots, M\}$ (different $u_i(x)$ corresponds to a different set of activation functions) and $r$, $q$, and $\tau$ are constants.

*Proof.* We first let $\boldsymbol{\eta} = (\eta_1, \eta_2, \ldots, \eta_M)$, $\widetilde{\boldsymbol{\eta}} = (\eta_1, \eta_2, \ldots, \eta_i - \tau, \ldots, \eta_M)$, and $\widehat{\boldsymbol{\eta}} = (\eta_1, \eta_2, \ldots, \eta_i - 2\tau, \ldots, \eta_M)$. The first moment of $v(x)$ is

$$E(v(x)) = r - q \int h(x)g(\boldsymbol{\eta})\exp\{\boldsymbol{\eta}^T u(x) - \tau u_i(x)\}\, dx$$

$$= r - q \int h(x)\frac{g(\boldsymbol{\eta})}{g(\widetilde{\boldsymbol{\eta}})} g(\widetilde{\boldsymbol{\eta}})\exp\{\widetilde{\boldsymbol{\eta}}^T u(x)\}\, dx = r - q\frac{g(\boldsymbol{\eta})}{g(\widetilde{\boldsymbol{\eta}})}.$$

Similarly the second moment can be computed as $E(v(x)^2) = r^2 + q^2 \frac{g(\boldsymbol{\eta})}{g(\widehat{\boldsymbol{\eta}})} - 2rq\frac{g(\boldsymbol{\eta})}{g(\widetilde{\boldsymbol{\eta}})}$. $\square$

A more detailed proof is provided in the supplementary material. With Theorem 1, what remains is to find the constants that make $v(x)$ strictly increasing and bounded (Table 1 shows some exponential-family distributions and their possible activation functions). For example in Equation (4), if $v(x) = r - q\exp(-\tau x)$, $\mathbf{a}_m = r - q(\frac{\mathbf{o}_d}{\mathbf{o}_d+\tau})^{\mathbf{o}_c}$ for the gamma distribution.

In the backpropagation, for distributions with two natural parameters the gradient consists of two terms. For example, $\frac{\partial E}{\partial \mathbf{o}_c} = \frac{\partial E}{\partial \mathbf{a}_m} \circ \frac{\partial \mathbf{a}_m}{\partial \mathbf{o}_c} + \frac{\partial E}{\partial \mathbf{a}_s} \circ \frac{\partial \mathbf{a}_s}{\partial \mathbf{o}_c}$, where $E$ is the error term of the network.

---

**Algorithm 1** Deep Nonlinear NPN

1: **Input:** Data $\mathcal{D} = \{(\mathbf{x}_i, \mathbf{y}_i)\}_{i=1}^N$, number of iterations $T$, learning rate $\rho_t$, number of layers $L$.
2: **for** $t = 1 : T$ **do**
3:     **for** $l = 1 : L$ **do**
4:         Apply Equation (1)-(4) to compute the *linear* and *nonlinear* transformation in layer $l$.
5:     **end for**
6:     Compute the *error* $E$ from $(\mathbf{o}_c^{(L)}, \mathbf{o}_d^{(L)})$ or $(\mathbf{a}_c^{(L)}, \mathbf{a}_d^{(L)})$.
7:     **for** $l = L : 1$ **do**
8:         Compute $\frac{\partial E}{\partial \mathbf{W}_m^{(l)}}$, $\frac{\partial E}{\partial \mathbf{W}_s^{(l)}}$, $\frac{\partial E}{\partial \mathbf{b}_m^{(l)}}$, and $\frac{\partial E}{\partial \mathbf{b}_s^{(l)}}$. Compute $\frac{\partial E}{\partial \mathbf{W}_c^{(l)}}$, $\frac{\partial E}{\partial \mathbf{W}_d^{(l)}}$, $\frac{\partial E}{\partial \mathbf{b}_c^{(l)}}$, and $\frac{\partial E}{\partial \mathbf{b}_d^{(l)}}$.
9:     **end for**
10:    Update $\mathbf{W}_c^{(l)}$, $\mathbf{W}_d^{(l)}$, $\mathbf{b}_c^{(l)}$, and $\mathbf{b}_d^{(l)}$ in all layers.
11: **end for**

---

### 2.4 Deep Nonlinear NPN

Naturally layers of nonlinear NPN can be stacked to form a deep NPN[1], as shown in Algorithm 1[2]. A deep NPN is in some sense similar to a PGM with a chain structure. Unlike PGM in general, however, NPN does not need costly inference algorithms like variational inference or Markov chain Monte Carlo. For some chain-structured PGM (e.g, hidden Markov models), efficient inference algorithms also exist due to their special structure. Similarly, the Markov property enables NPN to be efficiently trained in an end-to-end backpropagation learning fashion in the space of natural parameters.

PGM is known to be more flexible than NN in the sense that it can choose different distributions to depict different relationships among variables. A major drawback of PGM is its scalability especially

when the PGM is deep. Different from PGM, NN stacks relatively simple computational layers and learns the parameters using backpropagation, which is computationally more efficient than most algorithms for PGM. NPN has the potential to get the best of both worlds. In terms of flexibility, different types of exponential-family distributions can be chosen for the weights and neurons. Using gamma distributions for both the weights and neurons in NPN leads to a deep and nonlinear version of nonnegative matrix factorization [14] while an NPN with the Bernoulli distribution and sigmoid activation resembles a Bayesian treatment of sigmoid belief networks [17]. If Poisson distributions are chosen for the neurons, NPN becomes a neural analogue of deep Poisson factor analysis [26, 9].

Note that similar to the weight decay in NN, we may add *the KL divergence between the prior distributions and the learned distributions on the weights* to the error $E$ for regularization (we use isotropic Gaussian priors in the experiments). In NPN, the chosen prior distributions correspond to priors in Bayesian models and the learned distributions correspond to the approximation of posterior distributions on weights. Note that the generative story assumed here is that weights are sampled from the prior, and then output is generated (given all data) from these weights.

## 3 Variants of NPN

In this section, we introduce three NPN variants with different properties to demonstrate the flexibility and effectiveness of NPN. Note that in practice we use a transformed version of the natural parameters, referred to as *proxy natural parameters* here, instead of the original ones for computational efficiency. For example, in gamma distributions $p(x|c,d) = \Gamma(c)^{-1}d^c x^{c-1}\exp(-dx)$, we use proxy natural parameters $(c,d)$ during computation rather than the natural parameters $(c-1,-d)$.

### 3.1 Gamma NPN

The gamma distribution with support over positive values is an important member of the exponential family. The corresponding probability density function is $p(x|c,d) = \Gamma(c)^{-1}d^c x^{c-1}\exp(-dx)$ with $(c-1,-d)$ as its natural parameters (we use $(c,d)$ as proxy natural parameters). If we assume gamma distributions for $\mathbf{W}^{(l)}$, $\mathbf{b}^{(l)}$, $\mathbf{o}^{(l)}$, and $\mathbf{a}^{(l)}$, an AE formed by NPN becomes a deep and nonlinear version of nonnegative matrix factorization [14]. To see this, note that this AE with activation $v(x) = x$ and zero biases $\mathbf{b}^{(l)}$ is equivalent to finding a factorization of matrix $\mathbf{X}$ such that $\mathbf{X} = \mathbf{H}\prod_{l=\frac{L}{2}}^{L}\mathbf{W}^{(l)}$ where $\mathbf{H}$ denotes the middle-layer neurons and $\mathbf{W}^{(l)}$ has nonnegative entries from gamma distributions. In this gamma NPN, parameters $\mathbf{W}_c^{(l)}$, $\mathbf{W}_d^{(l)}$, $\mathbf{b}_c^{(l)}$, and $\mathbf{b}_d^{(l)}$ can be learned following Algorithm 1. We detail the algorithm as follows:

**Linear Transformation**: Since gamma distributions are assumed here, we can use the function $f(c,d) = (\frac{c}{d}, \frac{c}{d^2})$ to compute $(\mathbf{W}_m^{(l)}, \mathbf{W}_s^{(l)}) = f(\mathbf{W}_c^{(l)}, \mathbf{W}_d^{(l)})$, $(\mathbf{b}_m^{(l)}, \mathbf{b}_s^{(l)}) = f(\mathbf{b}_c^{(l)}, \mathbf{b}_d^{(l)})$, and $(\mathbf{o}_c^{(l)}, \mathbf{o}_d^{(l)}) = f^{-1}(\mathbf{o}_m^{(l)}, \mathbf{o}_s^{(l)})$ during the probabilistic linear transformation in Equation (1)-(3).

**Nonlinear Transformation**: With the proxy natural parameters for the gamma distributions over $\mathbf{o}^{(l)}$, the mean $\mathbf{a}_m^{(l)}$ and variance $\mathbf{a}_s^{(l)}$ for the nonlinearly transformed distribution over $\mathbf{a}^{(l)}$ would be obtained with Equation (4). Following Theorem 1, closed-form solutions are possible with $v(x) = r(1 - \exp(-\tau x))$ ($r = q$ and $u_i(x) = x$) where $r$ and $\tau$ are constants. Using this new activation function, we have (see Section 2.1 and 6.1 of the supplementary material for details on the function and derivation):

$$\mathbf{a}_m = \int p_o(\mathbf{o}|\mathbf{o}_c,\mathbf{o}_d)v(\mathbf{o})d\mathbf{o} = r(1 - \frac{\mathbf{o}_d^{\mathbf{o}_c}}{\Gamma(\mathbf{o}_c)} \circ \Gamma(\mathbf{o}_c) \circ (\mathbf{o}_d + \tau)^{-\mathbf{o}_c}) = r(1 - (\frac{\mathbf{o}_d}{\mathbf{o}_d + \tau})^{\mathbf{o}_c}),$$

$$\mathbf{a}_s = r^2((\frac{\mathbf{o}_d}{\mathbf{o}_d + 2\tau})^{\mathbf{o}_c} - (\frac{\mathbf{o}_d}{\mathbf{o}_d + \tau})^{2\mathbf{o}_c}).$$

**Error**: With $\mathbf{o}_c^{(L)}$ and $\mathbf{o}_d^{(L)}$, we can compute the regression error $E$ as the negative log-likelihood:

$$E = (\log\Gamma(\mathbf{o}_c^{(L)}) - \mathbf{o}_c^{(L)} \circ \log\mathbf{o}_d^{(L)} - (\mathbf{o}_c^{(L)} - 1) \circ \log\mathbf{y} + \mathbf{o}_d^{(L)} \circ \mathbf{y})\mathbf{1}^T,$$

where $\mathbf{y}$ is the observed output corresponding to $\mathbf{x}$. For classification, cross-entropy loss can be used as $E$. Following the computation flow above, BP can be used to learn $\mathbf{W}_c^{(l)}$, $\mathbf{W}_d^{(l)}$, $\mathbf{b}_c^{(l)}$, and $\mathbf{b}_d^{(l)}$.

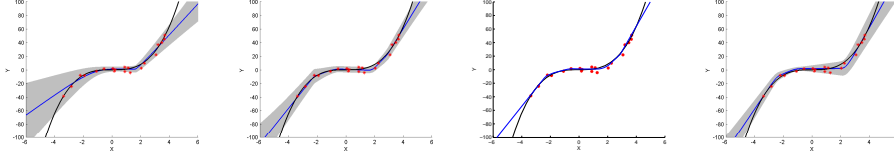

Figure 1: Predictive distributions for PBP, BDK, dropout NN, and NPN. The shaded regions correspond to $\pm 3$ standard deviations. The black curve is the data-generating function and blue curves show the mean of the predictive distributions. Red stars are the training data.

## 3.2 Gaussian NPN

Different from the gamma distribution which has support over positive values only, the Gaussian distribution, also an exponential-family distribution, can describe real-valued random variables. This makes it a natural choice for NPN. We refer to this NPN variant with Gaussian distributions over both the weights and neurons as Gaussian NPN. Details of Algorithm 1 for Gaussian NPN are as follows:

**Linear Transformation**: Besides support over real values, another property of Gaussian distributions is that the mean and variance can be used as proxy natural parameters, leading to an identity mapping function $f(c, d) = (c, d)$ which cuts the computation cost. We can use this function to compute $(\mathbf{W}_m^{(l)}, \mathbf{W}_s^{(l)}) = f(\mathbf{W}_c^{(l)}, \mathbf{W}_d^{(l)})$, $(\mathbf{b}_m^{(l)}, \mathbf{b}_s^{(l)}) = f(\mathbf{b}_c^{(l)}, \mathbf{b}_d^{(l)})$, and $(\mathbf{o}_c^{(l)}, \mathbf{o}_d^{(l)}) = f^{-1}(\mathbf{o}_m^{(l)}, \mathbf{o}_s^{(l)})$ during the probabilistic linear transformation in Equation (1)-(3).

**Nonlinear Transformation**: If the sigmoid activation $v(x) = \sigma(x) = \frac{1}{1+\exp(-x)}$ is used, $\mathbf{a}_m$ in Equation (4) would be (convolution of Gaussian with sigmoid is approximated by another sigmoid):

$$\mathbf{a}_m = \int \mathcal{N}(\mathbf{o}|\mathbf{o}_c, diag(\mathbf{o}_d)) \circ \sigma(\mathbf{o})d\mathbf{o} \approx \sigma(\frac{\mathbf{o}_c}{(1 + \zeta^2 \mathbf{o}_d)^{\frac{1}{2}}}), \tag{5}$$

$$\mathbf{a}_s = \int \mathcal{N}(\mathbf{o}|\mathbf{o}_c, diag(\mathbf{o}_d)) \circ \sigma(\mathbf{o})^2 d\mathbf{o} - \mathbf{a}_m^2 \approx \sigma(\frac{\alpha(\mathbf{o}_c + \beta)}{(1 + \zeta^2 \alpha^2 \mathbf{o}_d)^{1/2}}) - \mathbf{a}_m^2, \tag{6}$$

where $\alpha = 4 - 2\sqrt{2}$, $\beta = -\log(\sqrt{2} + 1)$, and $\zeta^2 = \pi/8$. Similar approximation can be applied for activation $v(x) = \tanh(x)$ since $\tanh(x) = 2\sigma(2x) - 1$.

If the ReLU activation $v(x) = \max(0, x)$ is used, we can use the techniques in [6] to obtain the first two moments of $\max(z_1, z_2)$ where $z_1$ and $z_2$ are Gaussian random variables. Full derivation for $v(x) = \sigma(x)$, $v(x) = \tanh(x)$, and $v(x) = \max(0, x)$ is left to the supplementary material.

**Error**: With $\mathbf{o}_c^{(L)}$ and $\mathbf{o}_d^{(L)}$ in the last layer, we can then compute the error $E$ as the KL divergence $\text{KL}(\mathcal{N}(\mathbf{o}_c^{(L)}, \text{diag}(\mathbf{o}_d^{(L)})) \| \mathcal{N}(\mathbf{y}_m, \text{diag}(\boldsymbol{\epsilon})))$, where $\boldsymbol{\epsilon}$ is a vector with all entries equal to a small value $\epsilon$. Hence the error $E = \frac{1}{2}(\frac{\boldsymbol{\epsilon}}{\mathbf{o}_d^{(L)}}\mathbf{1}^T + (\frac{1}{\mathbf{o}_d^{(L)}})(\mathbf{o}_c^{(L)} - \mathbf{y})^T - K + (\log \mathbf{o}_d^{(L)})\mathbf{1}^T - K \log \epsilon)$. For classification tasks, cross-entropy loss is used. Following the computation flow above, BP can be used to learn $\mathbf{W}_c^{(l)}$, $\mathbf{W}_d^{(l)}$, $\mathbf{b}_c^{(l)}$, and $\mathbf{b}_d^{(l)}$.

## 3.3 Poisson NPN

The Poisson distribution, as another member of the exponential family, is often used to model counts (e.g., counts of words, topics, or super topics in documents). Hence for text modeling, it is natural to assume Poisson distributions for neurons in NPN. Interestingly, this design of Poisson NPN can be seen as a neural analogue of some Poisson factor analysis models [26].

Besides closed-form nonlinear transformation, another challenge of Poisson NPN is to map the *pair* $(\mathbf{o}_m^{(l)}, \mathbf{o}_s^{(l)})$ to the *single* parameter $\mathbf{o}_c^{(l)}$ of Poisson distributions. According to the central limit theorem, we have $\mathbf{o}_c^{(l)} = \frac{1}{4}(2\mathbf{o}_m^{(l)} - 1 + \sqrt{(2\mathbf{o}_m^{(l)} - 1)^2 + 8\mathbf{o}_s^{(l)}})$ (see Section 3 and 6.3 of the supplementary material for proofs, justifications, and detailed derivation of Poisson NPN).

# 4 Experiments

In this section we evaluate variants of NPN and other state-of-the-art methods on four real-world datasets. We use Matlab (with GPU) to implement NPN, AE variants, and the 'vanilla' NN trained with dropout SGD (dropout NN). For other baselines, we use the Theano library [2] and MXNet [5].

Table 2: Test Error Rates on MNIST

| Method | BDK | BBB | Dropout1 | Dropout2 | gamma NPN | Gaussian NPN |
|--------|-----|-----|----------|----------|-----------|--------------|
| Error | 1.38% | 1.34% | 1.33% | 1.40% | 1.27% | **1.25%** |

Table 3: Test Error Rates for Different Size of Training Data

| Size | 100 | 500 | 2,000 | 10,000 |
|------|-----|-----|-------|--------|
| NPN | **29.97%** | **13.79%** | **7.89%** | **3.28%** |
| Dropout | 32.58% | 15.39% | 8.78% | 3.53% |
| BDK | 30.08% | 14.34% | 8.31% | 3.55% |

## 4.1 Toy Regression Task

To gain some insights into NPN, we start with a toy 1d regression task so that the predicted mean and variance can be visualized. Following [1], we generate 20 points in one dimension from a uniform distribution in the interval $[-4, 4]$. The target outputs are sampled from the function $y = x^3 + \epsilon_n$, where $\epsilon_n \sim \mathcal{N}(0, 9)$. We fit the data with the Gaussian NPN, BDK, and PBP (see the supplementary material for detailed hyperparameters). Figure 1 shows the predicted mean and variance of NPN, BDK, and PBP along with the mean provided by the dropout NN (for larger versions of figures please refer to the end of the supplementary materials). As we can see, the variance of PBP, BDK, and NPN diverges as $x$ is farther away from the training data. Both NPN's and BDK's predictive distributions are accurate enough to keep most of the $y = x^3$ curve inside the shaded regions with relatively low variance. An interesting observation is that the training data points become more scattered when $x > 0$. Ideally, the variance *should start diverging from $x = 0$*, which is what happens in NPN. However, PBP and BDK are *not sensitive enough* to capture this dispersion change. In another dataset, Boston Housing, the root mean square error for PBP, BDK, and NPN is 3.01, 2.82, and 2.57.

## 4.2 MNIST Classification

The MNIST digit dataset consists of 60,000 training images and 10,000 test images. All images are labeled as one of the 10 digits. We train the models with 50,000 images and use 10,000 images for validation. Networks with a structure of 784-800-800-10 are used for all methods, since 800 works best for the dropout NN (denoted as Dropout1 in Table 2) and BDK (BDK with a structure of 784-400-400-10 achieves an error rate of 1.41%). We also try the dropout NN with twice the number of hidden neurons (Dropout2 in Table 2) for fair comparison. For BBB, we directly quote their results from [4]. We implement BDK and NPN using the same hyperparameters as in [1] whenever possible. Gaussian priors are used for NPN (see the supplementary material for detailed hyperparameters).

As shown in Table 2, BDK and BBB achieve comparable performance with dropout NN (similar to [1], PBP is not included in the comparison since it supports regression only), and gamma NPN slightly outperforms dropout NN. Gaussian NPN is able to achieve a lower error rate of 1.25%. Note that BBB with Gaussian priors can only achieve an error rate of 1.82%; 1.34% is the result of using Gaussian mixture priors. For reference, the error rate for dropout NN with 1600 neurons in each hidden layer is 1.40%. The time cost per epoch is 18.3s, 16.2s, and 6.4s for NPN, BDK, NN respectively. Note that BDK is in C++ and NPN is in Matlab.

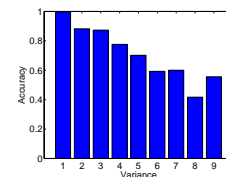

Figure 2: Classification accuracy for different variance (uncertainty). Note that '1' in the x-axis means $\mathbf{a}_s^{(L)}\mathbf{1}^T \in [0, 0.04)$, '2' means $\mathbf{a}_s^{(L)}\mathbf{1}^T \in [0.04, 0.08)$, etc.

To evaluate NPN's ability *as a Bayesian treatment to avoid overfitting*, we vary the size of the training set (from 100 to 10,000 data points) and compare the test error rates. As shown in Table 3, the margin between the Gaussian NPN and dropout NN increases as the training set shrinks. Besides, to verify the *effectiveness of the estimated uncertainty*, we split the test set into 9 subsets according NPN's estimated variance (uncertainty) $\mathbf{a}_s^{(L)}\mathbf{1}^T$ for each sample and show the accuracy for each subset in Figure 2. We can find that the more uncertain NPN is, the lower the accuracy, indicating that the estimated uncertainty is well calibrated.

## 4.3 Second-Order Representation Learning

Besides classification and regression, we also consider the problem of unsupervised representation learning with a subsequent link prediction task. Three real-world datasets, *Citeulike-a*, *Citeulike-t*, and *arXiv*, are used. The first two datasets are from [22, 23], collected separately from CiteULike in different ways to mimic different real-world settings. The third one is from arXiv as one of the SNAP datasets [15]. *Citeulike-a* consists of 16,980 documents, 8,000 terms, and 44,709 links (citations).

Table 4: Link Rank on Three Datasets

| Method | SAE | SDAE | VAE | gamma NPN | Gaussian NPN | Poisson NPN |
|--------|-----|------|-----|-----------|--------------|-------------|
| *Citeulike-a* | 1104.7 | 992.4 | 980.8 | 851.7 (935.8) | 750.6 (823.9) | **690.9 (5389.7)** |
| *Citeulike-t* | 2109.8 | 1356.8 | 1599.6 | 1342.3 (1400.7) | **1280.4 (1330.7)** | 1354.1 (9117.2) |
| *arXiv* | 4232.7 | 2916.1 | 3367.2 | 2796.4 (3038.8) | 2687.9 (2923.8) | **2684.1 (10791.3)** |

*Citeulike-t* consists of 25,975 documents, 20,000 terms, and 32,565 links. The last dataset, *arXiv*, consists of 27,770 documents, 8,000 terms, and 352,807 links.

The task is to perform unsupervised representation learning before feeding the extracted representations (middle-layer neurons) into a Bayesian LR algorithm [3]. We use the stacked autoencoder (SAE) [7], stacked denoising autoencoder (SDAE) [21], variational autoencoder (VAE) [13] as baselines (hyperparameters like weight decay and dropout rate are chosen by cross validation). As in SAE, we use different variants of NPN to form autoencoders where both the input and output targets are bag-of-words (BOW) vectors for the documents. The network structure for all models is $B$-100-50 ($B$ is the number of terms). Please refer to the supplementary material for detailed hyperparameters.

One major advantage of NPN over SAE and SDAE is that the learned representations are *distributions* instead of *point estimates*. Since representations from NPN contain both the *mean and variance*, we call them second-order representations. Note that although VAE also produces second-order representations, the variance part is simply parameterized by multilayer perceptrons while NPN's variance is naturally computed through propagation of distributions. These 50-dimensional representations with both mean and variance are fed into a Bayesian LR algorithm for link prediction (for deterministic AE the variance is set to 0).

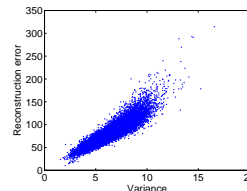

Figure 3: Reconstruction error and estimated uncertainty for each data point in *Citeulike-a*.

We use links among $80\%$ of the nodes (documents) to train the Bayesian LR and use other links as the test set. *link rank* and *AUC* (area under the ROC curve) are used as evaluation metrics. The link rank is the average rank of the observed links from test nodes to training nodes. We compute the AUC for every test node and report the average values. By definition, lower link rank and higher AUC indicate better predictive performance and imply more powerful representations.

Table 4 shows the link rank for different models. For fair comparison we also try all baselines with *double budget* (a structure of $B$-200-50) and report whichever has higher accuracy. As we can see, by treating representations as distributions rather than points in a vector space, NPN is able to achieve much lower link rank than all baselines, including VAE with variance information. The numbers in the brackets show the link rank of NPN if we discard the variance information. The performance gain from variance information *verifies the effectiveness of the variance (uncertainty) estimated by NPN*. Among different variants of NPN, the Gaussian NPN seems to perform better in datasets with fewer words like *Citeulike-t* (only $18.8$ words per document). The Poisson NPN, as a more natural choice to model text, achieves the best performance in datasets with more words (*Citeulike-a* and *arXiv*). The performance in AUC is consistent with that in terms of the link rank (see Section 4 of the supplementary material). To further verify the effectiveness of the estimated uncertainty, we plot the reconstruction error and the variance $\mathbf{o}_s^{(L)}\mathbf{1}^T$ for each data point of *Citeulike-a* in Figure 3. As we can see, higher uncertainty often indicates not only *higher reconstruction error $E$* but also *higher variance in $E$*.

## 5   Conclusion

We have introduced a family of models, called natural-parameter networks, as a novel class of probabilistic NN to combine the merits of NN and PGM. NPN regards the weights and neurons as arbitrary exponential-family distributions rather than just point estimates or factorized Gaussian distributions. Such flexibility enables richer descriptions of hierarchical relationships among latent variables and adds another degree of freedom to customize NN for different types of data. Efficient sampling-free backpropagation-compatible algorithms are designed for the learning of NPN. Experiments show that NPN achieves state-of-the-art performance on classification, regression, and representation learning tasks. As possible extensions of NPN, it would be interesting to connect NPN to arbitrary PGM to form fully Bayesian deep learning models [24, 25], allowing even richer descriptions of relationships among latent variables. It is also worth noting that NPN cannot be defined as generative models and, unlike PGM, the same NPN model cannot be used to support multiple types of inference (with different observed and hidden variables). We will try to address these limitations in our future work.

## Footnotes

[1]Although the approximation accuracy may decrease as NPN gets deeper during feedforward computation, it can be automatically adjusted according to data during backpropagation.

[2]Note that since the first part of Equation (1) and the last part of Equation (4) are canceled out, we can directly use $(\mathbf{a}_m^{(l)}, \mathbf{a}_s^{(l)})$ without computing $(\mathbf{a}_c^{(l)}, \mathbf{a}_d^{(l)})$ here.

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
