[Supplementary Material]

# Supplementary Materials for Natural-Parameter Networks: A Class of Probabilistic Neural Networks

**Hao Wang, Xingjian Shi, Dit-Yan Yeung**
Hong Kong University of Science and Technology
{hwangaz,xshiab,dyyeung}@cse.ust.hk

## 1 Proof of Theorem 1

**Theorem 1.** *Assume an exponential-family distribution $p_o(x|\boldsymbol{\eta}) = h(x)g(\boldsymbol{\eta})\exp\{\boldsymbol{\eta}^T u(x)\}$, where the vector $u(x) = (u_1(x), u_2(x), \ldots, u_M(x))^T$ (M is the number of natural parameters). If activation function $v(x) = r - q\exp(-\tau u_i(x))$ is used, the first two moments of $v(x)$, $\int p_o(x|\boldsymbol{\eta})v(x)dx$ and $\int p_o(x|\boldsymbol{\eta})v(x)^2 dx$, can be expressed in closed form. Here $i \in \{1, 2, \ldots, M\}$ and $r$, $q$, and $\tau$ are constants.*

*Proof.* We first let $\boldsymbol{\eta} = (\eta_1, \eta_2, \ldots, \eta_M)$, $\widetilde{\boldsymbol{\eta}} = (\eta_1, \eta_2, \ldots, \eta_i - \tau, \ldots, \eta_M)$, and $\widehat{\boldsymbol{\eta}} = (\eta_1, \eta_2, \ldots, \eta_i - 2\tau, \ldots, \eta_M)$. The first moment of $v(x)$ is

$$
\begin{aligned}
E(v(x)) &= \int p_o(x|\boldsymbol{\eta})(r - q\exp(-\tau u_i(x)))dx \\
&= r - q\int h(x)g(\boldsymbol{\eta})\exp\{\boldsymbol{\eta}^T u(x) - \tau u_i(x)\}dx \\
&= r - q\int h(x)\frac{g(\boldsymbol{\eta})}{g(\widetilde{\boldsymbol{\eta}})}g(\widetilde{\boldsymbol{\eta}})\exp\{\widetilde{\boldsymbol{\eta}}^T u(x)\}dx \\
&= r - q\frac{g(\boldsymbol{\eta})}{g(\widetilde{\boldsymbol{\eta}})}\int h(x)g(\widetilde{\boldsymbol{\eta}})\exp\{\widetilde{\boldsymbol{\eta}}^T u(x)\}dx \\
&= r - q\frac{g(\boldsymbol{\eta})}{g(\widetilde{\boldsymbol{\eta}})}.
\end{aligned}
$$

Figure 1: Activation functions for the gamma distribution (left), the beta distribution (middle), and the Rayleigh distribution (right).

Similarly the second moment

$$
\begin{aligned}
E(v(x)^2) &= \int p_o(x|\boldsymbol{\eta})(r - q\exp(-\tau u_i(x)))^2 dx \\
&= \int p_o(x|\boldsymbol{\eta})(r^2 + q^2\exp(-2\tau u_i(x)) - 2rq\exp(-\tau u_i(x)))dx \\
&= r^2 \int p_o(x|\boldsymbol{\eta})dx + q^2 \int h(x)g(\boldsymbol{\eta})\exp\{\boldsymbol{\eta}^T u(x) - 2\tau u_i(x)\}dx \\
&\quad - 2rq \int h(x)g(\boldsymbol{\eta})\exp\{\boldsymbol{\eta}^T u(x) - \tau u_i(x)\}dx \\
&= r^2 + q^2 \int h(x)g(\boldsymbol{\eta})\exp\{\widehat{\boldsymbol{\eta}}^T u(x)\}dx - 2rq \int h(x)g(\boldsymbol{\eta})\exp\{\widetilde{\boldsymbol{\eta}}^T u(x)\}dx \\
&= r^2 + q^2 \int h(x)\frac{g(\boldsymbol{\eta})}{g(\widehat{\boldsymbol{\eta}})}g(\widehat{\boldsymbol{\eta}})\exp\{\widetilde{\boldsymbol{\eta}}^T u(x)\}dx - 2rq \int h(x)\frac{g(\boldsymbol{\eta})}{g(\widetilde{\boldsymbol{\eta}})}g(\widetilde{\boldsymbol{\eta}})\exp\{\widetilde{\boldsymbol{\eta}}^T u(x)\}dx \\
&= r^2 + q^2\frac{g(\boldsymbol{\eta})}{g(\widehat{\boldsymbol{\eta}})}\int h(x)g(\widehat{\boldsymbol{\eta}})\exp\{\widetilde{\boldsymbol{\eta}}^T u(x)\}dx - 2rq\frac{g(\boldsymbol{\eta})}{g(\widetilde{\boldsymbol{\eta}})}\int h(x)g(\widetilde{\boldsymbol{\eta}})\exp\{\widetilde{\boldsymbol{\eta}}^T u(x)\}dx \\
&= r^2 + q^2\frac{g(\boldsymbol{\eta})}{g(\widehat{\boldsymbol{\eta}})} - 2rq\frac{g(\boldsymbol{\eta})}{g(\widetilde{\boldsymbol{\eta}})}.
\end{aligned}
$$

$\square$

## 2 Exponential-Family Distributions and Activation Functions

In this section we provide a list of exponential-family distributions with corresponding activation functions that could lead to close-form expressions of the first two moments of $v(x)$, namely $E(v(x))$ and $E(v(x)^2)$. With Theorem 1, we only need to find the constants ($r$, $q$, and $\tau$) that make $v(x) = r - q\exp(-\tau u_i(x))$ monotonically increasing and bounded.

As mentioned in the paper, we use the activation function $v(x) = r(1 - \exp(-\tau x))$ for the gamma NPN and the Poisson NPN. Figure 1(left) plots this function with different $\tau$ when $r = 1$. As we can see, this function has a similar shape with the positive half of $v(x) = \tanh(x)$ (the negative part is irrelevant because both the gamma distribution and the Poisson distribution have support over positive values only). Note that the activation function $v(x) = 1 - \exp(-1.5x)$ is very similar to $v(x) = \tanh(x)$.

For beta distributions, since the support set is $(0, 1)$ the domain of the activation function is also $(0, 1)$. In this case $v(x) = qx^\tau$ is a reasonable activation function when $\tau \in (0, 1)$ and $q = 1$. Figure 1(middle) shows this function with differnt $\tau$ when $q = 1$. Since we expect the nonlinearly transformed distribution to be another beta distribution, the domain of the function should be $(0, 1)$ and the field should be $[0, 1]$. With these criteria, $v(x) = 1.3\tanh(x)$ might be a better activation function than $v(x) = \tanh(x)$. As shown in the figure, different $\tau$ leads to different shapes of the function.

For Rayleigh distributions with support over positive reals, $v(x) = r - qe^{-\tau x^2}$ is a proper activation function with the domain $x \in \mathbb{R}^+$. Figure 1(right) plots this function with different $\tau$ when $r = q = 1$. We can see that this function also has a similar shape with the positive half of $v(x) = \tanh(x)$.

## 2.1 Gamma Distributions

For gamma distributions with $(v(x) = r(1 - \exp(-\tau x))$, as mentioned in the paper,

$$
\begin{aligned}
\mathbf{a}_m &= \int p_o(\mathbf{o}_j|\mathbf{o}_c, \mathbf{o}_d) v(\mathbf{o}) d\mathbf{o} = r \int_0^{+\infty} \frac{1}{\Gamma(\mathbf{o}_c)} \mathbf{o}_d^{\mathbf{o}_c} \circ \mathbf{o}^{\mathbf{o}_c - 1} e^{-\mathbf{o}_d \circ \mathbf{o}} (1 - e^{-\tau \mathbf{o}}) d\mathbf{o} \\
&= r(1 - \frac{\mathbf{o}_d^{\mathbf{o}_c}}{\Gamma(\mathbf{o}_c)} \int_0^{+\infty} \mathbf{o}^{\mathbf{o}_c - 1} e^{-(\mathbf{o}_d + \tau) \circ \mathbf{o}} d\mathbf{o}) \\
&= r(1 - \frac{\mathbf{o}_d^{\mathbf{o}_c}}{\Gamma(\mathbf{o}_c)} \circ \Gamma(\mathbf{o}_c) \circ (\mathbf{o}_d + \tau)^{-\mathbf{o}_c}) \\
&= r(1 - (\frac{\mathbf{o}_d}{\mathbf{o}_d + \tau})^{\mathbf{o}_c}).
\end{aligned}
$$

Similarly we have

$$
\begin{aligned}
\mathbf{a}_s &= \int p_o(\mathbf{o}_j|\mathbf{o}_c, \mathbf{o}_d) v(\mathbf{o})^2 d\mathbf{o} - \mathbf{a}_m^2 \\
&= r^2 \int_0^{+\infty} \frac{1}{\Gamma(\mathbf{o}_c)} \mathbf{o}_d^{\mathbf{o}_c} \circ \mathbf{o}^{\mathbf{o}_c - 1} e^{-\mathbf{o}_d \circ \mathbf{o}} (1 - 2e^{-\tau \mathbf{o}} + e^{-2\tau \mathbf{o}}) d\mathbf{o} - \mathbf{a}_m^2 \\
&= r^2 (1 - 2\frac{\mathbf{o}_d^{\mathbf{o}_c}}{\Gamma(\mathbf{o}_c)} \circ \Gamma(\mathbf{o}_c) \circ (\mathbf{o}_d + \tau)^{-\mathbf{o}_c} + \frac{\mathbf{o}_d^{\mathbf{o}_c}}{\Gamma(\mathbf{o}_c)} \circ \Gamma(\mathbf{o}_c) \circ (\mathbf{o}_d + 2\tau)^{-\mathbf{o}_c}) - \mathbf{a}_m^2 \\
&= r^2 (1 - 2(\frac{\mathbf{o}_d}{\mathbf{o}_d + \tau})^{\mathbf{o}_c} + (\frac{\mathbf{o}_d}{\mathbf{o}_d + 2\tau})^{\mathbf{o}_c}) - \mathbf{a}_m^2 \\
&= r^2 ((\frac{\mathbf{o}_d}{\mathbf{o}_d + 2\tau})^{\mathbf{o}_c} - (\frac{\mathbf{o}_d}{\mathbf{o}_d + \tau})^{2\mathbf{o}_c}).
\end{aligned}
$$

Equivalently we can obtain the same $\mathbf{a}_m$ and $\mathbf{a}_s$ by following Theorem 1. For the gamma distribution

$$
p(x|c, d) = \frac{d^c}{\Gamma(c)} \exp\{(c - 1)\log x + (-b)x\}.
$$

Thus we have $\boldsymbol{\eta} = (c - 1, -d)^T$, $u(x) = (\log x, x)^T$, and $g(\boldsymbol{\eta}) = \frac{d^c}{\Gamma(c)}$. Using $v(x) = r(1 - \exp(-\tau x))$ implies $g(\widetilde{\boldsymbol{\eta}}) = \frac{(d + \tau)^c}{\Gamma(c)}$ and $g(\widehat{\boldsymbol{\eta}}) = \frac{(d + 2\tau)^c}{\Gamma(c)}$. Hence we have

$$
\mathbf{a}_m = r - r\frac{g(\boldsymbol{\eta})}{g(\widetilde{\boldsymbol{\eta}})} = r(1 - (\frac{\mathbf{o}_d}{\mathbf{o}_d + \tau})^{\mathbf{o}_c}),
$$

and the variance

$$
\begin{aligned}
\mathbf{a}_s &= r^2 + r^2 \frac{g(\boldsymbol{\eta})}{g(\widehat{\boldsymbol{\eta}})} - 2r^2 \frac{g(\boldsymbol{\eta})}{g(\widetilde{\boldsymbol{\eta}})} - r^2 (1 - \frac{g(\boldsymbol{\eta})}{g(\widetilde{\boldsymbol{\eta}})})^2 \\
&= r^2 ((\frac{\mathbf{o}_d}{\mathbf{o}_d + 2\tau})^{\mathbf{o}_c} - (\frac{\mathbf{o}_d}{\mathbf{o}_d + \tau})^{2\mathbf{o}_c}).
\end{aligned}
$$

## 2.2 Poisson Distributions

For Poisson distributions with $v(x) = r(1 - \exp(-\tau x))$, using the Taylor expansion of $\exp(\exp(-\tau)\lambda)$ with respect to $\lambda$,

$$
\exp(\exp(-\tau)\lambda) = \sum_{x=0}^{+\infty} \frac{\lambda^x \exp(-\tau x)}{x!},
$$

we have

$$\mathbf{a}_m = r \sum_{x=0}^{+\infty} \frac{\mathbf{o}_c^x \exp(-\mathbf{o}_c)}{x!}(1 - \exp(-\tau x))$$

$$= r\left(\sum_{x=0}^{+\infty} \frac{\mathbf{o}_c^x \exp(-\mathbf{o}_c)}{x!} - \sum_{x=0}^{+\infty} \frac{\mathbf{o}_c^x \exp(-\mathbf{o}_c)}{x!} \exp(-\tau x)\right)$$

$$= r(1 - \exp(-\mathbf{o}_c) \sum_{x=0}^{+\infty} \frac{\mathbf{o}_c^x \exp(-\tau x)}{x!})$$

$$= r(1 - \exp(-\mathbf{o}_c) \exp(\exp(-\tau)\mathbf{o}_c))$$

$$= r(1 - \exp((\exp(-\tau) - 1)\mathbf{o}_c)).$$

Similarly, we have

$$\mathbf{a}_s = r^2 \sum_{x=0}^{+\infty} \frac{\mathbf{o}_c^x \exp(-\mathbf{o}_c)}{x!}(1 - \exp(-\tau x))^2 - \mathbf{a}_m^2$$

$$= r^2 \sum_{x=0}^{+\infty} \frac{\mathbf{o}_c^x \exp(-\mathbf{o}_c)}{x!}(1 - 2\exp(-\tau x) + \exp(-2\tau x)) - \mathbf{a}_m^2$$

$$= r^2\left(\sum_{x=0}^{+\infty} \frac{\mathbf{o}_c^x \exp(-\mathbf{o}_c)}{x!} - 2\sum_{x=0}^{+\infty} \frac{\mathbf{o}_c^x \exp(-\mathbf{o}_c)}{x!}\exp(-\tau x) + \sum_{x=0}^{+\infty} \frac{\mathbf{o}_c^x \exp(-\mathbf{o}_c)}{x!}\exp(-2\tau x)\right) - \mathbf{a}_m^2$$

$$= r^2(\exp((\exp(-2\tau) - 1)\mathbf{o}_c) - \exp(2(\exp(-\tau) - 1)\mathbf{o}_c).$$

Equivalently we can follow Theorem 1 to obtain $\mathbf{a}_m$ and $\mathbf{a}_s$. For the Poisson distribution

$$p(x|c) = \frac{1}{x!}\exp(-c)\exp\{x \log c\}$$

Thus we have $\boldsymbol{\eta} = \log c$, $u(x) = x$, and $g(\boldsymbol{\eta}) = \exp(-c)$. Using $v(x) = r(1 - \exp(-\tau x))$ implies $g(\widetilde{\boldsymbol{\eta}}) = \exp(-\exp(-\tau)c)$ and $g(\widehat{\boldsymbol{\eta}}) = \exp(-\exp(-2\tau)c)$. Hence we have

$$\mathbf{a}_m = r - r\frac{g(\boldsymbol{\eta})}{g(\widetilde{\boldsymbol{\eta}})}$$

$$= r(1 - \exp((\exp(-\tau) - 1)\mathbf{o}_c)),$$

and the variance

$$\mathbf{a}_s = r^2 + r^2\frac{g(\boldsymbol{\eta})}{g(\widehat{\boldsymbol{\eta}})} - 2r^2\frac{g(\boldsymbol{\eta})}{g(\widetilde{\boldsymbol{\eta}})} - r^2(1 - \frac{g(\boldsymbol{\eta})}{g(\widetilde{\boldsymbol{\eta}})})^2$$

$$= r^2(\exp((\exp(-2\tau) - 1)\mathbf{o}_c) - \exp(2(\exp(-\tau) - 1)\mathbf{o}_c)).$$

## 2.3 Gaussian Distributions

In this subsection, we provide detailed derivation of $(\mathbf{a}_m, \mathbf{a}_s)$ for Gaussian distributions.

### 2.3.1 Sigmoid Activation

We start by proving the following theorem:

**Theorem 2.** *Consider a univariate Gaussian distribution $\mathcal{N}(x|\mu, \sigma^2)$ and the probit function $\Phi(x) = \int_{-\infty}^{x} \mathcal{N}(\theta|0, 1)d\theta$. If $\zeta^2 = \frac{\pi}{8}$, for any constants $a$ and $b$, the following equation holds:*

$$\int \Phi(\zeta a(x + b)\mathcal{N}(x|\mu, \sigma^2)dx = \Phi\left(\frac{\zeta a(\mu + b)}{(1 + \zeta^2 a^2 \sigma^2)^{\frac{1}{2}}}\right). \tag{1}$$

*Proof.* Making the change of variable $x = \mu + \sigma z$, we have

$$\mathcal{I} = \int \Phi(\zeta a(x + b)\mathcal{N}(x|\mu, \sigma^2)dx$$

$$= \int \Phi(\zeta a(\mu + \sigma z + b))\frac{1}{(2\pi\sigma)^{\frac{1}{2}}}\exp\{-\frac{1}{2}z^2\}\sigma dz.$$

Taking the derivative with respect to $\mu$,

$$\frac{\partial \mathcal{I}}{\partial \mu} = \frac{\zeta a}{2\pi} \int \exp\{-\frac{1}{2}z^2 - \frac{1}{2}\zeta^2 a^2(\mu + \sigma z + b)^2\}dz$$

$$= \frac{\zeta a}{2\pi} \int \exp\{-\frac{1}{2}z^2 - \frac{1}{2}\zeta^2 a^2(\mu^2 + \sigma^2 z^2 + b^2 + 2\mu\sigma z + 2\mu b + 2\sigma zb)\}dz$$

$$= \frac{\zeta a}{2\pi} \int \exp\{-\frac{1}{2}(1 + \zeta^2 a^2\sigma^2)(z^2 + \frac{2\zeta^2 a^2\sigma(\mu + b)}{1 + \zeta^2 a^2\sigma^2}z + \frac{(\mu^2 + b^2 + 2\mu b)\zeta^2 a^2}{1 + \zeta^2 a^2\sigma^2})\}dz$$

$$= \frac{\zeta a}{2\pi} \int \exp\{-\frac{1}{2}(1 + \zeta^2 a^2\sigma^2)((z + \frac{\zeta^2 a^2\sigma(\mu + b)}{1 + \zeta^2 a^2\sigma^2})^2 - \frac{\zeta^4 a^4\sigma^2(\mu + b)^2}{(1 + \zeta^2 a^2\sigma^2)^2} + \frac{(\mu + b)^2\zeta^2 a^2}{1 + \zeta^2 a^2\sigma^2})\}dz$$

$$= \frac{\zeta a}{2\pi} \int \exp\{-\frac{1}{2}(1 + \zeta^2 a^2\sigma^2)((z + \frac{\zeta^2 a^2\sigma(\mu + b)}{1 + \zeta^2 a^2\sigma^2})^2 + \frac{1}{2}\frac{\zeta^4 a^4\sigma^2(\mu + b)^2}{1 + \zeta^2 a^2\sigma^2} - \frac{1}{2}(\mu + b)^2\zeta^2 a^2)\}dz$$

$$= \frac{\zeta a}{2\pi} \int \exp\{-\frac{1}{2}(1 + \zeta^2 a^2\sigma^2)((z + \frac{\zeta^2 a^2\sigma(\mu + b)}{1 + \zeta^2 a^2\sigma^2})^2 - \frac{1}{2}\frac{(\mu + b)^2\zeta^2 a^2}{1 + \zeta^2 a^2\sigma^2})\}dz$$

$$= \frac{1}{(2\pi)^{\frac{1}{2}}}\frac{\zeta a}{(1 + \zeta^2 a^2\sigma^2)^{\frac{1}{2}}} \exp\{-\frac{1}{2}\frac{(\mu + b)^2\zeta^2 a^2}{1 + \zeta^2 a^2\sigma^2}\}.$$

Taking derivative of the right-hand side of Equation (1) also gives

$$\frac{1}{(2\pi)^{\frac{1}{2}}}\frac{\zeta a}{(1 + \zeta^2 a^2\sigma^2)^{\frac{1}{2}}} \exp\{-\frac{1}{2}\frac{(\mu + b)^2\zeta^2 a^2}{1 + \zeta^2 a^2\sigma^2}\},$$

which means the derivatives of the left and right hand sides of Equation (1) with respect to $\mu$ are equal. When $\mu$ approaches negative infinity, the derivatives go to zero, which implies that the constant of the integration is zero. Hence Equation (1) holds. $\square$

As mentioned in the paper (with a slight abuse of notation on $\sigma$), if the sigmoid activation $v(x) = \sigma(x) = \frac{1}{1+\exp(-x)}$ is used,

$$\mathbf{a}_m = \int \mathcal{N}(\mathbf{o}|\mathbf{o}_c, diag(\mathbf{o}_d)) \circ \frac{1}{1 + \exp(-\mathbf{o})} d\mathbf{o}$$

$$\approx \int \mathcal{N}(\mathbf{o}|\mathbf{o}_c, diag(\mathbf{o}_d)) \circ \Phi(\zeta\mathbf{o}) d\mathbf{o}. \tag{2}$$

Following Theorem 2 with $a = 1$ and $b = 0$, we have

$$\mathbf{a}_m \approx \Phi(\frac{\mathbf{o}_c}{(\zeta^{-2} + \mathbf{o}_d)^{\frac{1}{2}}})$$

$$= \sigma(\frac{\mathbf{o}_c}{(1 + \zeta^2\mathbf{o}_d)^{\frac{1}{2}}}).$$

For the variance,

$$\mathbf{a}_s \approx \int \mathcal{N}(\mathbf{o}|\mathbf{o}_c, diag(\mathbf{o}_d)) \circ \Phi(\zeta\alpha(\mathbf{o} + \beta)) d\mathbf{o} - \mathbf{a}_m^2$$

$$= \sigma(\frac{\alpha(\mathbf{o}_m + \beta)}{(1 + \zeta^2\alpha^2\mathbf{o}_s)^{1/2}}) - \mathbf{a}_m^2. \tag{3}$$

Equation (3) holds due to Theorem 2 with $a = \alpha = 4 - 2\sqrt{2}$ and $b = \beta = -\log(\sqrt{2} + 1)$.

### 2.3.2 Hyperbolic Tangent Activation

If the tanh activation $v(x) = \tanh(x)$ is used, since $\tanh(x) = 2\sigma(2x) - 1$, we have

$$
\begin{aligned}
\mathbf{a}_m &= \int \mathcal{N}(\mathbf{o}|\mathbf{o}_c, diag(\mathbf{o}_d)) \circ (2\sigma(2\mathbf{o}) - 1)d\mathbf{o} \\
&= 2 \int \mathcal{N}(\mathbf{o}|\mathbf{o}_c, diag(\mathbf{o}_d)) \circ \sigma(2\mathbf{o})d\mathbf{o} - 1 \\
&\approx 2 \int \mathcal{N}(\mathbf{o}|\mathbf{o}_c, diag(\mathbf{o}_d)) \circ \Phi(2\zeta\mathbf{o})d\mathbf{o} - 1 \\
&= 2\Phi\left(\frac{2\zeta\mathbf{o}_c}{(1 + 4\zeta^2\mathbf{o}_d)^{\frac{1}{2}}}\right) - 1 \\
&= 2\sigma\left(\frac{\mathbf{o}_c}{(0.25 + \zeta^2\mathbf{o}_d)^{\frac{1}{2}}}\right) - 1,
\end{aligned}
\tag{4}
$$

where Equation (4) is due to Theorem 2 with $a = 2$ and $b = 0$. For the variance of $\mathbf{a}$:

$$
\begin{aligned}
\mathbf{a}_s &= \int \mathcal{N}(\mathbf{o}|\mathbf{o}_c, diag(\mathbf{o}_d)) \circ (2\sigma(2\mathbf{o}) - 1)^2 d\mathbf{o} - \mathbf{a}_m^2 \\
&= \int \mathcal{N}(\mathbf{o}|\mathbf{o}_c, diag(\mathbf{o}_d)) \circ (4\sigma(2\mathbf{o})^2 - 4\sigma(2\mathbf{o}) + 1)d\mathbf{o} - \mathbf{a}_m^2 \\
&\approx \int \mathcal{N}(\mathbf{o}|\mathbf{o}_c, diag(\mathbf{o}_d)) \circ (4\Phi(\zeta\alpha(\mathbf{o} + \beta)) - 4\sigma(2\mathbf{o}) + 1)d\mathbf{o} - \mathbf{a}_m^2 \\
&= 4\sigma\left(\frac{\alpha(\mathbf{o}_c + \beta)}{(1 + \zeta^2\alpha^2\mathbf{o}_d)^{\frac{1}{2}}}\right) - \mathbf{a}_m^2 - 2\mathbf{a}_m - 1,
\end{aligned}
\tag{5}
$$

where Equation (5) follows from Theorem 2 with $a = \alpha = 8 - 4\sqrt{2}$ and $b = \beta = -0.5\log(\sqrt{2} + 1)$.

### 2.3.3 ReLU Activation

If the ReLU activation $v(x) = \max(0, x)$ is used, we can use the techniques in [3] to obtain the first two moments of $z = \max(z_1, z_2)$ where $z_1 \sim \mathcal{N}(\mu_1, \sigma_1^2)$ and $z_2 \sim \mathcal{N}(\mu_2, \sigma_2^2)$. Specifically,

$$
\begin{aligned}
E(z) &= \mu_1\Phi(\gamma) + \mu_2\Phi(-\gamma) + \nu\phi(\gamma) \\
E(z^2) &= (\mu_1^2 + \sigma_1^2)\Phi(\gamma) + (\mu_2^2 + \sigma_2^2)\Phi(-\gamma) + (\mu_1 + \mu_2)\nu\phi(\gamma),
\end{aligned}
$$

where $\Phi(x) = \int_{-\infty}^{x} \mathcal{N}(\theta|0, 1)d\theta$, $\phi(x) = \mathcal{N}(x|0, 1)$, $\nu = \sqrt{\sigma_1^2 + \sigma_2^2}$, and $\gamma = \frac{\mu_1 - \mu_2}{\nu}$. If $\mathcal{N}(\mu_1, \sigma_1^2) = \mathcal{N}(c, d)$ and $\mathcal{N}(\mu_2, \sigma_2^2) = \mathcal{N}(0, 0)$, we recover the probabilistic version of ReLU. In this case,

$$
\begin{aligned}
E(z) &= \Phi\left(\frac{c}{\sqrt{d}}\right)c + \sqrt{\frac{d}{2\pi}}\exp\left\{-\frac{1}{2}\frac{c^2}{d}\right\} \\
D(z) &= E(z^2) - E(z)^2 = \Phi\left(\frac{c}{\sqrt{d}}\right)(c^2 + d) + \frac{c\sqrt{d}}{\sqrt{2\pi}}\exp\left\{-\frac{1}{2}\frac{c^2}{d}\right\} - c^2.
\end{aligned}
$$

Hence we have the following equations as in the main text:

$$
\begin{aligned}
\mathbf{a}_m^{(l)} &= \Phi\left(\mathbf{o}_m \circ \mathbf{o}_s^{(l)}{}^{-\frac{1}{2}}\right) \circ \mathbf{o}_m + \frac{\sqrt{\mathbf{o}_s}}{\sqrt{2\pi}} \circ \exp\left(-\frac{\mathbf{o}_m{}^2}{2\mathbf{o}_s}\right) \\
\mathbf{a}_s^{(l)} &= \Phi\left(\mathbf{o}_m \circ \mathbf{o}_s^{(l)}{}^{-\frac{1}{2}}\right) \circ (\mathbf{o}_m{}^2 + \mathbf{o}_s) + \frac{\mathbf{o}_m^{(l)} \circ \sqrt{\mathbf{o}_s}}{\sqrt{2\pi}} \circ \exp\left(-\frac{\mathbf{o}_m{}^2}{2\mathbf{o}_s}\right) - \mathbf{a}_m^2.
\end{aligned}
$$

## 3 Mapping Function for Poisson Distributions

Since the mapping function involves Gaussian approximation to a Poisson distribution, we start with proving the connection between Gaussian distributions and Poisson distributions.

**Lemma 1.** *Assume $Y$ is a Poisson random variable with mean $c$ and variance $c$. If $X_1, X_2, \ldots, X_c$ are independent Poisson random variables with mean $1$, we have:*

$$Y = \sum_{i=1}^{c} X_i$$

*Proof.* We can use the concept of moment generating functions (i.e., two distributions are identical if they have exactly the same moment generating function), which is defined as $M(t) = E(\exp(tZ))$ for a random variable $Z$, to prove the lemma. The moment generating function for a Poisson random variable with mean $c$ and variance $c$ is:

$$M_1(t) = \exp(c(\exp(t) - 1)).$$

On the other hand, the moment generating function for $\sum_{i=1}^{c} X_i$ is:

$$
\begin{aligned}
M_2(t) &= E(\exp(t \sum_{i=1}^{c} X_i)) \\
&= E(\prod_{i=1}^{c} \exp(tX_i)) \\
&= \prod_{i=1}^{c} E(\exp(tX_i)) \quad (6) \\
&= \prod_{i=1}^{c} \exp(\exp(t) - 1) \quad (7) \\
&= \exp(c(\exp(t) - 1)) \\
&= M_1(t),
\end{aligned}
$$

where Equation (6) is due to the fact that $X_1, X_2, \ldots, X_c$ are independent. Equation (7) is the result of using the moment generating functions of Poisson distributions. Since $\sum_{i=1}^{c} X_i$ has exactly the same moment generating function as a Poisson random variable with mean $c$ and variance $c$, by definition of $Y$, we have:

$$Y = \sum_{i=1}^{c} X_i$$

□

**Theorem 3.** *A Poisson distribution with mean $c$ and variance $c$ can be approximated by a Gaussian distribution $\mathcal{N}(c, c)$ if $c$ is sufficiently large.*

*Proof.* We first use $Y$ to denote the random variable corresponding to the Poisson distribution with mean $c$ and variance $c$. According to Lemma 1, we have $Y = \sum_{i=1}^{c} X_i$ where $X_1, X_2, \ldots, X_c$ are independent Poisson random variables with mean $1$. Hence,

$$
\begin{aligned}
\frac{Y - c}{\sqrt{c}} &= \frac{\sum_{i=1}^{c} X_i - c}{\sqrt{c}} \\
&= \sqrt{c}(\frac{1}{c} \sum_{i=1}^{c} X_i - 1),
\end{aligned}
$$

where $\frac{1}{c} \sum_{i=1}^{c} X_i$ is the sample mean. By the central limit theorem, we know that if $c$ is sufficiently large, $\sqrt{c}(\frac{1}{c} \sum_{i=1}^{c} X_i - 1)$ can be approximated by the Gaussian distribution $\mathcal{N}(0, 1)$. Thus $Y$ can be approximated by the Gaussian distribution $\mathcal{N}(c, c)$.

□

Note that although $c$ is a nonnegative integer above, the proof can be easily generalized to the case in which $c$ is a nonnegative real value.

During the feedforward computation of the Poisson NPN, after obtaining the mean $\mathbf{o}_m^{(l)}$ and variance $\mathbf{o}_s^{(l)}$ of the linearly transformed distribution over $\mathbf{o}^{(l)}$, we map them back to the proxy natural parameters $\mathbf{o}_c^{(l)}$. Unfortunately the mean and variance of a Poisson are the same, which is obviously not the case for $\mathbf{o}_m^{(l)}$ and $\mathbf{o}_s^{(l)}$. Here we propose to find $\mathbf{o}_c^{(l)}$ by minimizing the KL divergence of the factorized Poisson distribution $p(\mathbf{o}^{(l)}|\mathbf{o}_c^{(l)})$ and the Gaussian distribution $\mathcal{N}(\mathbf{o}_m^{(l)}, diag(\mathbf{o}_s^{(l)}))$[1].

Since the direct KL divergence involves the computation of an infinite series in the entropy term of the Poisson distribution, closed-form solutions are not available. To address the problem, we use a Gaussian distribution $\mathcal{N}(\mathbf{o}_c^{(l)}, diag(\mathbf{o}_c^{(l)}))$ as a proxy of the Poisson distribution with the mean $\mathbf{o}_c^{(l)}$ (which is justified by Theorem 3)[2]. Specifically, we aim to find a Gaussian distribution $\mathcal{N}(\mathbf{o}_c^{(l)}, diag(\mathbf{o}_c^{(l)}))$ to best approximate $\mathcal{N}(\mathbf{o}_m^{(l)}, diag(\mathbf{o}_s^{(l)}))$ and directly use $\mathbf{o}_c^{(l)}$ in the new Gaussian as the result of mapping.

For simplicity, we consider the univariate case where we aim to find a Gaussian distribution $\mathcal{N}(c, c)$ to approximate $\mathcal{N}(m, s)$. The KL divergence between $\mathcal{N}(c, c)$ and $\mathcal{N}(m, s)$

$$D_{KL}(\mathcal{N}(c,c)\|\mathcal{N}(m,s)) = \frac{1}{2}\left(\frac{c}{s} + \frac{(c-m)^2}{s} - 1 + \log s - \log c\right),$$

which is convex with respect to $c > 0$. We set the gradient of $D_{KL}(\mathcal{N}(c,c)\|\mathcal{N}(m,s))$ with respect to $c$ as 0 and solve for $c$, giving

$$c = \frac{2m - 1 \pm \sqrt{(2m-1)^2 + 8s}}{4}.$$

Since in Poisson distributions, $c$ is always positive, there is only one solution for $c$:

$$c = \frac{2m - 1 + \sqrt{(2m-1)^2 + 8s}}{4}.$$

Thus the mapping is

$$\mathbf{o}_c^{(l)} = \frac{1}{4}\left(2\mathbf{o}_m^{(l)} - 1 + \sqrt{(2\mathbf{o}_m^{(l)} - 1)^2 + 8\mathbf{o}_s^{(l)}}\right).$$

## 4  AUC for Link Prediction and Different Data Splitting

In this section, we show the AUC for different models on the link prediction task. As we can see in Table 1 above, the result in AUC is consistent with that in link rank (as shown in Table 3 of the paper). NPN is able to achieve much higher AUC than SAE, SDAE, and VAE. Among different variants of NPN, the Gaussian NPN seems to perform better in datasets with fewer words like *Citeulike-t* (18.8 words per document). The Poisson NPN, as a more natural choice to model text, achieves the best performance in datasets with more words (*Citeulike-a* with 66.6 words per document and *arXiv* with 88.8 words per document).

For the link prediction task, we also try to split the data in a different way and compare the performance of different models. Specifically, we randomly select $80\%$ of the *observed links* (rather than nodes) as the training set and use the others as the test set. The results are consistent with those for the original data-splitting method.

## 5  Hyperparameters and Preprocessing

In this section we provide details on the hyperparameters and preprocessing of the experiments as mentioned in the paper.

Table 1: AUC on Three Datasets

| Method | SAE | SDAE | VAE | gamma NPN | Gaussian NPN | Poisson NPN |
|---|---|---|---|---|---|---|
| *Citeulike-a* | 0.915 | 0.917 | 0.929 | 0.938 | 0.951 | **0.956** |
| *Citeulike-t* | 0.891 | 0.920 | 0.922 | 0.936 | **0.940** | 0.934 |
| *arXiv* | 0.811 | 0.840 | 0.834 | 0.861 | 0.878 | **0.879** |

## 5.1 Toy Regression Task

For the toy 1d regression task, we use networks with one hidden layer containing 100 neurons and ReLU activation, as in [1, 4]. For the Gaussian NPN, we use the KL divergence loss and isotropic Gaussian priors with precision $10^{-4}$ for the weights (and biases). The same priors are used in other experiments.

## 5.2 MNIST Classification

For preprocessing, following [2, 1], pixel values are normalized to the range $[0, 1]$. For the NPN variants, we use these hyperparameters: minibatch size 128, number of epochs 2000 (the same as BDK). For the learning rate, AdaDelta is used. Note that since NPN is dropout-compatible, we can use dropout (with nearly no additional cost) for effective regularization. The training and testing of dropout NPN are similar to those of the vanilla dropout NN.

## 5.3 Second-Order Representation Learning

For all models, we preprocess the BOW vectors by normalizing them into the range $[0, 1]$. Although theoretically Poisson NPN does not need any preprocessing since Poisson distributions naturally model word counts, in practice, we find normalizing the BOW vectors will increase both stability during training and the predictive performance. For simplicity, in the Poisson NPN, $r$ is set to 1 and $\tau = 0.1$ (these two hyperparameters can be tuned to further improve performance). For the Gaussian NPN, sigmoid activation is used. The other hyperparameters of NPN are the same as in the MNIST experiments.

---

**Algorithm 1** Deep Nonlinear NPN

---

1: **Input:** The data $\{(\mathbf{x}_i, \mathbf{y}_i)\}_{i=1}^{N}$, number of iterations $T$, learning rate $\rho_t$, number of layers $L$.
2: **for** $t = 1 : T$ **do**
3:     **for** $l = 1 : L$ **do**
4:         Compute $(\mathbf{o}_m^{(l)}, \mathbf{o}_s^{(l)})$ from $(\mathbf{a}_m^{(l-1)}, \mathbf{a}_s^{(l-1)})$. $(\mathbf{o}_c^{(l)}, \mathbf{o}_d^{(l)}) = f^{-1}(\mathbf{o}_m^{(l)}, \mathbf{o}_s^{(l)})$.
5:         Compute $(\mathbf{a}_m^{(l)}, \mathbf{a}_s^{(l)})$ from $(\mathbf{o}_c^{(l)}, \mathbf{o}_d^{(l)})$.
6:     **end for**
7:     Compute the error $E$.
8:     **for** $l = L : 1$ **do**
9:         Compute $\frac{\partial E}{\partial \mathbf{W}_m^{(l)}}, \frac{\partial E}{\partial \mathbf{W}_s^{(l)}}, \frac{\partial E}{\partial \mathbf{b}_m^{(l)}}$, and $\frac{\partial E}{\partial \mathbf{b}_s^{(l)}}$. Compute $\frac{\partial E}{\partial \mathbf{W}_c^{(l)}}, \frac{\partial E}{\partial \mathbf{W}_d^{(l)}}, \frac{\partial E}{\partial \mathbf{b}_c^{(l)}}$, and $\frac{\partial E}{\partial \mathbf{b}_d^{(l)}}$.
10:     **end for**
11:     Update $\mathbf{W}_c^{(l)}, \mathbf{W}_d^{(l)}, \mathbf{b}_c^{(l)}$, and $\mathbf{b}_d^{(l)}$ in all layers.
12: **end for**

---

# 6 Details on Variants of NPN

## 6.1 Gamma NPN

In gamma NPN, parameters $\mathbf{W}_c^{(l)}$, $\mathbf{W}_d^{(l)}$, $\mathbf{b}_c^{(l)}$, and $\mathbf{b}_d^{(l)}$ can be learned following Algorithm 1. Specifically, during the feedforward phase, we will compute the error $E$ given the input $\mathbf{a}_m^{(0)} = \mathbf{x}$ ($\mathbf{a}_s^{(0)} = \mathbf{0}$) and the parameters ($\mathbf{W}_c^{(l)}$, $\mathbf{W}_d^{(l)}$, $\mathbf{b}_c^{(l)}$, and $\mathbf{b}_d^{(l)}$). With the mean $\mathbf{a}_m^{(l-1)}$ and variance $\mathbf{a}_s^{(l-1)}$ from the previous layer, $\mathbf{o}_m^{(l)}$ and $\mathbf{o}_s^{(l)}$ can be computed according to equations in Section 2.2 of the paper, where

$$(\mathbf{W}_m^{(l)}, \mathbf{W}_s^{(l)}) = (\mathbf{W}_c^{(l)} \circ \mathbf{W}_d^{(l)-1}, \mathbf{W}_c^{(l)} \circ \mathbf{W}_d^{(l)-2}), \ (\mathbf{b}_m^{(l)}, \mathbf{b}_s^{(l)}) = (\mathbf{b}_c^{(l)} \circ \mathbf{b}_d^{(l)-1}, \mathbf{b}_c^{(l)} \circ \mathbf{b}_d^{(l)-2}). \quad (8)$$

After that we can get the proxy natural parameters using $(\mathbf{o}_c^{(l)}, \mathbf{o}_d^{(l)}) = (\mathbf{o}_m^{(l)} \circ \mathbf{o}_s^{(l)}{}^{-1}, \mathbf{o}_m^{(l)}{}^2 \circ \mathbf{o}_s^{(l)})$.

With the proxy natural parameters for the gamma distributions over $\mathbf{o}^{(l)}$, the mean $\mathbf{a}_m^{(l)}$ and variance $\mathbf{a}_s^{(l)}$ for the nonlinearly transformed distribution over $\mathbf{a}^{(l)}$ would be obtained. As mentioned before, using traditional activation functions like tanh $v(x) = \tanh(x)$ and ReLU $v(x) = \max(0, x)$ could not give us closed-form solutions for the integrals. Following Theorem 1, closed-form solutions are possible with $v(x) = r(1 - \exp(-\tau x))$ ($r = q$ and $u_i(x) = x$) where $r$ and $\tau$ are constants. This function has a similar shape with the positive half of $v(x) = \tanh(x)$ with $r$ as the saturation point and $\tau$ controlling the slope.

With the computation procedure for the feedforward phase, the gradients of $E$ with respect to parameters $\mathbf{W}_c^{(l)}$, $\mathbf{W}_d^{(l)}$, $\mathbf{b}_c^{(l)}$, and $\mathbf{b}_d^{(l)}$ can be derived and used for backpropagation. Note that to ensure positive entries in the parameters we can use the function $k(x) = \log(1 + \exp(x))$ or $k(x) = \exp(x - h)$. For example, we can let $\mathbf{b}_c^{(l)} = \log(1 + \exp(\mathbf{b}_{c'}^{(l)}))$ and treat $\mathbf{b}_{c'}^{(l)}$ as parameters to learn instead of $\mathbf{b}_c^{(l)}$.

We can add the KL divergence between the learned distribution and the prior distribution on weights to the objective function to regularize gamma NPN. If we use an isotropic Gaussian prior $\mathcal{N}(0, \lambda_s^{-1})$ for each entry of the weights, we can compute the KL divergence for each entry (between $Gam(c, d)$ and $\mathcal{N}(0, \lambda_s^{-1})$) as:

$$KL(Gam(x|c, d)\|\mathcal{N}(x|0, \lambda_s^{-1}))$$
$$= \int Gam(x|c, d) \log Gam(x|c, d)dx - \int Gam(x|c, d) \log \mathcal{N}(x|0, \lambda_s^{-1})$$
$$= -\log \Gamma(c) + (c - 1)\psi(c) + \log d - c + \frac{1}{2}\log(2\pi) - \frac{1}{2}\log \lambda_s + \frac{1}{2}\lambda_s \int \frac{d^c}{\Gamma(c)} x^{c+2-1} \exp(-dx)dx$$
$$= -\log \Gamma(c) + (c - 1)\psi(c) + \log d - c + \frac{1}{2}\log(2\pi) - \frac{1}{2}\log \lambda_s + \frac{1}{2}\lambda_s \frac{\Gamma(c + 2)}{\Gamma(c)}$$
$$= -\log \Gamma(c) + (c - 1)\psi(c) + \log d - c + \frac{1}{2}\log(2\pi) - \frac{1}{2}\log \lambda_s + \frac{1}{2}\lambda_s c(c + 1), \tag{9}$$

where $\psi(x) = \frac{d}{dx} \log \Gamma(x)$ is the digamma function.

## 6.2 Gaussian NPN

For details on the Bayesian nonlinear transformation, please refer to Section 2.3 above. For the KL divergence between the learned distribution and the prior distribution on weights, we can compute it as:

$$KL(\mathcal{N}(x|c, d)\|\mathcal{N}(x|0, \lambda_s^{-1})) = \frac{1}{2}(\lambda_s + \lambda_s c^2 - 1 - \log \lambda_s - \log d), \tag{10}$$

As we can see, the term $-\frac{1}{2}\log d$ will help to prevent the learned variance $d$ from collapsing to $0$ (in practice we can use $\frac{1}{2}\lambda_d(d - h)^2$, where $\lambda_d$ and $h$ are hyperparameters, to approximate this term for better numerical stability) and the term $\frac{1}{2}c^2$ is equivalent to L2 regularization. Similar to BDK, we can use a mixture of Gaussians as the prior distribution.

## 6.3 Poisson NPN

The Poisson distribution, as another member of the exponential family, is often used to model counts (e.g., number of events happened or number of words in a document). Different from the previous distributions, it has support over nonnegative integers. The Poisson distribution takes the form $p(x|c) = \frac{c^x \exp(-c)}{x!}$ with one single natural parameter $\log c$ (we use $c$ as the proxy natural parameter). It is this single natural parameter that makes the learning of a Poisson NPN trickier. For text modeling, assuming Poisson distributions for neurons is natural because they can model the counts of words and topics (even super topics) in documents. Here we assume a factorized Poisson distribution $p(\mathbf{o}^{(l)}|\mathbf{o}_c^{(l)}) = \prod_j p(\mathbf{o}_j^{(l)}|\mathbf{o}_{c,j}^{(l)})$ and do the same for $\mathbf{a}^{(l)}$. To ensure having positive natural parameters we use gamma distributions for the weights. Interestingly, this design of Poisson NPN can be seen as a neural analogue of some Poisson factor analysis models [5].

Following Algorithm 1, we need to compute $E$ during the feedforward phase given the input $\mathbf{a}_m^{(0)} = \mathbf{x}$ ($\mathbf{a}_s^{(0)} = \mathbf{0}$) and the parameters ($\mathbf{W}_c^{(l)}$, $\mathbf{W}_d^{(l)}$, $\mathbf{b}_c^{(l)}$, and $\mathbf{b}_d^{(l)}$), the first step being to compute the mean $\mathbf{o}_m^{(l)}$ and variance $\mathbf{o}_s^{(l)}$. Since gamma distributions are assumed for the weights, we can compute the mean and variance of the weights as follows:

$$(\mathbf{W}_m^{(l)}, \mathbf{W}_s^{(l)}) = (\mathbf{W}_c^{(l)} \circ \mathbf{W}_d^{(l)^{-1}}, \mathbf{W}_c^{(l)} \circ \mathbf{W}_d^{(l)^{-2}}), \quad (\mathbf{b}_m^{(l)}, \mathbf{b}_s^{(l)}) = (\mathbf{b}_c^{(l)} \circ \mathbf{b}_d^{(l)^{-1}}, \mathbf{b}_c^{(l)} \circ \mathbf{b}_d^{(l)^{-2}}). \quad (11)$$

Having computed the mean $\mathbf{o}_m^{(l)}$ and variance $\mathbf{o}_s^{(l)}$ of the linearly transformed distribution over $\mathbf{o}^{(l)}$, we map them back to the proxy natural parameters $\mathbf{o}_c^{(l)}$. Unfortunately the mean and variance of a Poisson are the same, which is obviously not the case for $\mathbf{o}_m^{(l)}$ and $\mathbf{o}_s^{(l)}$. Hence we propose to find $\mathbf{o}_c^{(l)}$ by minimizing the KL divergence of the factorized Poisson distribution $p(\mathbf{o}^{(l)}|\mathbf{o}_c^{(l)})$ and the Gaussian distribution $\mathcal{N}(\mathbf{o}_m^{(l)}, diag(\mathbf{o}_s^{(l)}))$, resulting in the mapping (see Section 3 for proofs and justifications):

$$\mathbf{o}_c^{(l)} = \frac{1}{4}(2\mathbf{o}_m^{(l)} - 1 + \sqrt{(2\mathbf{o}_m^{(l)} - 1)^2 + 8\mathbf{o}_s^{(l)}}). \quad (12)$$

After finding $\mathbf{o}_c^{(l)}$, the next step in Algorithm 1 is to get the mean $\mathbf{a}_m^{(l)}$ and variance $\mathbf{a}_s^{(l)}$ of the nonlinearly transformed distribution. As is the case for gamma NPN, traditional activation functions will not give us closed-form solutions. Fortunately, the activation $v(x) = r(1 - \exp(-\tau x))$ also works for Poisson NPN. Specifically,

$$\mathbf{a}_m = r \sum_{x=0}^{+\infty} \frac{\mathbf{o}_c^x \exp(-\mathbf{o}_c)}{x!}(1 - \exp(-\tau x)) = r(1 - \exp((\exp(-\tau) - 1)\mathbf{o}_c)),$$

where the superscript $(l)$ is dropped. Similarly, we have

$$\mathbf{a}_s = r^2(\exp((\exp(-2\tau) - 1)\mathbf{o}_c) - \exp(2(\exp(-\tau) - 1)\mathbf{o}_c)).$$

Full derivation is provided in Section 2.2.

Once we go through $L$ layers to get the proxy natural parameters $\mathbf{o}_c^{(L)}$ for the distribution over $\mathbf{o}^{(L)}$, the error $E$ can be computed as the negative log-likelihood. Assuming that the target output $\mathbf{y}$ has nonnegative integers as entries,

$$E = -\mathbf{1}^T(\mathbf{y} \circ \log \mathbf{o}_c^{(L)} - \mathbf{o}_c^{(L)} - \log(\mathbf{y}!)).$$

For $\mathbf{y}$ with real-valued entries, the L2 loss could be used as the error $E$. Note that if we use the normalized BOW as the target output, the same error $E$ can be used as the Gaussian NPN. Besides this loss term, we can add the KL divergence term in Equation (9) to regularize Poisson NPN.

During backpropagation, the gradients are computed to update the parameters $\mathbf{W}_c^{(l)}$, $\mathbf{W}_d^{(l)}$, $\mathbf{b}_c^{(l)}$, and $\mathbf{b}_d^{(l)}$. Interestingly, since $\mathbf{o}_c^{(l)}$ is guaranteed to be nonnegative, the model still works even if we directly use $\mathbf{W}_m^{(l)}$ and $\mathbf{W}_s^{(l)}$ as parameters, though the resulting models are not exactly the same. In the experiments, we use this Poisson NPN for a Bayesian autoencoder and feed the extracted second-order representations into a Bayesian LR algorithm for link prediction.

# 7 Derivation of Gradients

In this section we list the gradients used in backpropagation to update the NPN parameters.

## 7.1 Gamma NPN

In the following we assume an activation function of $v(x) = r(1 - \exp(-\tau x))$ and use $\psi(x) = \frac{d}{dx}\log\Gamma(x)$ to denote the *digamma* function. $E$ is the error we want to minimize.

$E \rightarrow \mathbf{o}^{(L)}$:

$$\frac{\partial E}{\partial \mathbf{o}_c^{(L)}} = \psi(\mathbf{o}_c^{(L)}) - \log \mathbf{o}_d^{(L)} - \log \mathbf{y}$$

$$\frac{\partial E}{\partial \mathbf{o}_d^{(L)}} = -\frac{\mathbf{o}_c^{(L)}}{\mathbf{o}_d^{(L)}} + \mathbf{y}.$$

$\mathbf{o}^{(l)} \to \mathbf{a}^{(l-1)}$:

$$\frac{\partial E}{\partial \mathbf{a}_m^{(l-1)}} = \frac{\partial E}{\partial \mathbf{o}_m^{(l)}} \mathbf{W}_m^{(l)\,T} + (\frac{\partial E}{\partial \mathbf{o}_s^{(l)}} \mathbf{W}_s^{(l)\,T}) \circ 2\mathbf{a}_m^{(l-1)}$$

$$\frac{\partial E}{\partial \mathbf{a}_s^{(l-1)}} = \frac{\partial E}{\partial \mathbf{o}_s^{(l)}} \mathbf{W}_s^{(l)\,T} + \frac{\partial E}{\partial \mathbf{o}_s^{(l)}} (\mathbf{W}_m^{(l)} \circ \mathbf{W}_m^{(l)})^T$$

$\mathbf{a}^{(l)} \to \mathbf{o}^{(l)}$:

$$\frac{\partial E}{\partial \mathbf{o}_c^{(l)}} = \frac{\partial E}{\partial \mathbf{a}_m^{(l)}} \circ (-r(\frac{\mathbf{o}_d^{(l)}}{\mathbf{o}_d^{(l)} + \tau})^{\mathbf{o}_c^{(l)}} \circ \log(\frac{\mathbf{o}_d^{(l)}}{\mathbf{o}_d^{(l)} + \tau}))$$

$$+ r^2 \frac{\partial E}{\partial \mathbf{a}_s^{(l)}} ((\frac{\mathbf{o}_d^{(l)}}{\mathbf{o}_d^{(l)} + 2\tau})^{\mathbf{o}_c^{(l)}} \circ \log(\frac{\mathbf{o}_d^{(l)}}{\mathbf{o}_d^{(l)} + 2\tau}) - 2(\frac{\mathbf{o}_d^{(l)}}{\mathbf{o}_d^{(l)} + 2\tau})^{2\mathbf{o}_c^{(l)}} \circ \log(\frac{\mathbf{o}_d^{(l)}}{\mathbf{o}_d^{(l)} + 2\tau}))$$

$$\frac{\partial E}{\partial \mathbf{o}_c^{(l)}} = \frac{\partial E}{\partial \mathbf{a}_m^{(l)}} \circ (-r\mathbf{o}_c^{(l)} \circ (\frac{\mathbf{o}_d^{(l)}}{\mathbf{o}_d^{(l)} + \tau})^{\mathbf{o}_c^{(l)}-1} \circ \frac{\tau}{(\mathbf{o}_d^{(l)} + \tau)^2})$$

$$+ r^2 \frac{\partial E}{\partial \mathbf{a}_s^{(l)}} \circ (\mathbf{o}_c^{(l)} \circ (\frac{\mathbf{o}_d^{(l)}}{\mathbf{o}_d^{(l)} + 2\tau})^{\mathbf{o}_c^{(l)}-1} \circ \frac{2\tau}{(\mathbf{o}_d^{(l)} + 2\tau)^2})$$

$$- 2\mathbf{o}_c^{(l)} \circ (\frac{\mathbf{o}_d^{(l)}}{\mathbf{o}_d^{(l)} + \tau})^{2\mathbf{o}_c^{(l)}-1} \circ \frac{\tau}{(\mathbf{o}_d^{(l)} + \tau)^2}).$$

$\mathbf{o}^{(l)} \to \mathbf{W}^{(l)}, \mathbf{o}^{(l)} \to \mathbf{b}^{(l)}$:

The gradients with respect to the mean-variance pairs:

$$\frac{\partial E}{\partial \mathbf{W}_m^{(l)}} = \mathbf{a}_m^{(l-1)\,T} \frac{\partial E}{\partial \mathbf{o}_m^{(l)}} + (\mathbf{a}_s^{(l-1)\,T} \frac{\partial E}{\partial \mathbf{o}_s^{(l)}}) \circ 2\mathbf{W}_m^{(l)}$$

$$\frac{\partial E}{\partial \mathbf{W}_s^{(l)}} = \mathbf{a}_s^{(l-1)\,T} \frac{\partial E}{\partial \mathbf{o}_s^{(l)}} + (\mathbf{a}_m^{(l-1)} \circ \mathbf{a}_m^{(l-1)})^T \frac{\partial E}{\partial \mathbf{o}_s^{(l)}}$$

$$\frac{\partial E}{\partial \mathbf{b}_m^{(l)}} = \frac{\partial E}{\partial \mathbf{o}_m^{(l)}}$$

$$\frac{\partial E}{\partial \mathbf{b}_s^{(l)}} = \frac{\partial E}{\partial \mathbf{o}_s^{(l)}}$$

The gradients with respect to the proxy natural parameters:

$$\frac{\partial E}{\partial \mathbf{W}_c^{(l)}} = \frac{\partial E}{\partial \mathbf{W}_m^{(l)}} \circ \frac{1}{\mathbf{W}_d^{(l)}} + \frac{\partial E}{\partial \mathbf{W}_s^{(l)}} \circ \frac{1}{\mathbf{W}_d^{(l)2}}$$

$$\frac{\partial E}{\partial \mathbf{W}_d^{(l)}} = -\frac{\partial E}{\partial \mathbf{W}_m^{(l)}} \circ \frac{\mathbf{W}_c^{(l)}}{\mathbf{W}_d^{(l)2}} - 2\frac{\partial E}{\partial \mathbf{W}_s^{(l)}} \circ \frac{\mathbf{W}_c^{(l)}}{\mathbf{W}_d^{(l)3}}$$

$$\frac{\partial E}{\partial \mathbf{b}_c^{(l)}} = \frac{\partial E}{\partial \mathbf{b}_m^{(l)}} \circ \frac{1}{\mathbf{b}_d^{(l)}} + \frac{\partial E}{\partial \mathbf{b}_s^{(l)}} \circ \frac{1}{\mathbf{b}_d^{(l)2}}$$

$$\frac{\partial E}{\partial \mathbf{b}_d^{(l)}} = -\frac{\partial E}{\partial \mathbf{b}_m^{(l)}} \circ \frac{\mathbf{b}_c^{(l)}}{\mathbf{b}_d^{(l)2}} - 2\frac{\partial E}{\partial \mathbf{b}_s^{(l)}} \circ \frac{\mathbf{b}_c^{(l)}}{\mathbf{b}_d^{(l)3}}$$

## 7.2   Gaussian NPN

In the following we assume the sigmoid activation function and use cross-entropy loss. Other activation functions and loss could be derived similarly. For the equations below, $\alpha = 4 - 2\sqrt{2}$, $\beta = -\log(\sqrt{2}+1)$, $\zeta^2 = \frac{\pi}{8}$, and $\kappa(x) = (1 + \zeta^2 x)^{-\frac{1}{2}}$.

$E \to \mathbf{o}^{(L)}$:

$$\frac{\partial E}{\partial \mathbf{o}_m^{(L)}} = (\sigma(\kappa(\mathbf{o}_s^{(L)}) \circ \mathbf{o}_m^{(L)}) - \mathbf{y}) \circ \kappa(\mathbf{o}_s^{(L)})$$

$$\frac{\partial E}{\partial \mathbf{o}_s^{(L)}} = (\sigma(\kappa(\mathbf{o}_s^{(L)}) \circ \mathbf{o}_m^{(L)}) - \mathbf{y}) \circ \mathbf{o}_m^{(L)} \circ (-\frac{\pi}{16}(1 + \pi \mathbf{o}_s^{(L)}/8)^{-3/2}).$$

$\mathbf{o}^{(l)} \to \mathbf{a}^{(l-1)}$:

$$\frac{\partial E}{\partial \mathbf{a}_m^{(l-1)}} = \frac{\partial E}{\partial \mathbf{o}_m^{(l)}} \mathbf{W}_m^{(l)T} + (\frac{\partial E}{\partial \mathbf{o}_s^{(l)}} \mathbf{W}_s^{(l)T}) \circ 2\mathbf{a}_m^{(l-1)}$$

$$\frac{\partial E}{\partial \mathbf{a}_s^{(l-1)}} = \frac{\partial E}{\partial \mathbf{o}_s^{(l)}} \mathbf{W}_s^{(l)T} + \frac{\partial E}{\partial \mathbf{o}_s^{(l)}} (\mathbf{W}_m^{(l)} \circ \mathbf{W}_m^{(l)})^T.$$

$\mathbf{a}^{(l)} \to \mathbf{o}^{(l)}$:

$$\frac{\partial E}{\partial \mathbf{o}_m^{(l)}} = \frac{\partial E}{\partial \mathbf{a}_m^{(l)}} \circ dsigmoid(\kappa(\mathbf{o}_s^{(l)}) \circ \mathbf{o}_m^{(l)}) \circ \kappa(\mathbf{o}_s^{(l)})$$

$$+ \alpha \frac{\partial E}{\partial \mathbf{a}_s^{(l)}} \circ dsigmoid(\frac{\alpha(\mathbf{o}_m^{(l)} + \beta)}{(1 + \zeta^2 \alpha^2 \mathbf{o}_s^{(l)})^{1/2}}) \circ (1 + \zeta^2 \alpha^2 \mathbf{o}_s^{(l)})^{-1/2}$$

$$- 2\mathbf{a}_m^{(l)} \circ \frac{\partial E}{\partial \mathbf{a}_s^{(l)}} \circ dsigmoid(\kappa(\mathbf{o}_s^{(l)}) \circ \mathbf{o}_m^{(l)}) \circ \kappa(\mathbf{o}_s^{(l)})$$

$$\frac{\partial E}{\partial \mathbf{o}_s^{(l)}} = \frac{\partial E}{\partial \mathbf{a}_m^{(l)}} \circ dsigmoid(\kappa(\mathbf{o}_s^{(l)}) \circ \mathbf{o}_m^{(l)}) \circ \mathbf{o}_m^{(l)} \circ (-\frac{1}{2}\zeta^2(1 + \zeta^2 \mathbf{o}_s^{(l)})^{-3/2})$$

$$+ \frac{\partial E}{\partial \mathbf{a}_s^{(l)}} \circ dsigmoid(\frac{\alpha(\mathbf{o}_m^{(l)} + \beta)}{(1 + \zeta^2 \alpha^2 \mathbf{o}_s^{(l)})^{1/2}}) \circ (\alpha(\mathbf{o}_m^{(l)} + \beta)) \circ (-\frac{1}{2}\zeta^2 \alpha^2 (1 + \zeta^2 \alpha^2 \mathbf{o}_s^{(l)})^{-3/2})$$

$$- 2\mathbf{a}_m^{(l)} \circ \frac{\partial E}{\partial \mathbf{a}_s^{(l)}} \circ dsigmoid(\kappa(\mathbf{o}_s^{(l)}) \circ \mathbf{o}_m^{(l)}) \circ \mathbf{o}_m^{(l)} \circ (-\frac{1}{2}\zeta^2(1 + \zeta^2 \mathbf{o}_s^{(l)})^{-3/2}),$$

where $dsigmoid(x)$ is the gradient of $\sigma(x)$.

$\mathbf{o}^{(l)} \to \mathbf{W}^{(l)}, \mathbf{o}^{(l)} \to \mathbf{b}^{(l)}$:

$$\frac{\partial E}{\partial \mathbf{W}_c^{(l)}} = \mathbf{a}_m^{(l-1)T} \frac{\partial E}{\partial \mathbf{o}_m^{(l)}} + (\mathbf{a}_s^{(l-1)T} \frac{\partial E}{\partial \mathbf{o}_s^{(l)}}) \circ 2\mathbf{W}_c^{(l)}$$

$$\frac{\partial E}{\partial \mathbf{W}_d^{(l)}} = \mathbf{a}_s^{(l-1)T} \frac{\partial E}{\partial \mathbf{o}_s^{(l)}} + (\mathbf{a}_m^{(l-1)} \circ \mathbf{a}_m^{(l-1)})^T \frac{\partial E}{\partial \mathbf{o}_s^{(l)}}$$

$$\frac{\partial E}{\partial \mathbf{b}_c^{(l)}} = \frac{\partial E}{\partial \mathbf{o}_m^{(l)}}$$

$$\frac{\partial E}{\partial \mathbf{b}_d^{(l)}} = \frac{\partial E}{\partial \mathbf{o}_s^{(l)}}$$

Note that we directly use the mean and variance as proxy natural parameters here.

### 7.3 Poisson NPN

In the following we assume the activation function $v(x) = r(1 - \exp(\tau x))$ and use Poisson regression loss $E = \mathbf{1}^T(\mathbf{y} \circ \log \mathbf{o}_c^{(L)} - \mathbf{o}_c^{(L)} - \log(\mathbf{y}!))$ (the target output $\mathbf{y}$ is a vector with nonnegative integer entries). Gamma distributions are used on weights.

$E \to \mathbf{o}_c^{(L)}$:

$$\frac{\partial E}{\partial \mathbf{o}_c^{(L)}} = \frac{\mathbf{y}}{\mathbf{o}_c^{(L)}} - 1.$$

$\mathbf{o}_c^{(l)} \to \mathbf{o}_m^{(l)}, \mathbf{o}_c^{(l)} \to \mathbf{o}_s^{(l)}$:

$$\frac{\partial E}{\partial \mathbf{o}_m^{(l)}} = \frac{\partial E}{\mathbf{o}_c^{(l)}} \circ (\frac{1}{2} + \frac{1}{2}((2\mathbf{o}_m^{(l)} - 1)^2 + 8\mathbf{o}_s^{(l)})^{-\frac{1}{2}} \circ (2\mathbf{o}_m^{(l)} - 1))$$

$$\frac{\partial E}{\partial \mathbf{o}_s^{(l)}} = \frac{\partial E}{\mathbf{o}_c^{(l)}} \circ ((2\mathbf{o}_m^{(l)} - 1)^2 + 8\mathbf{o}_s^{(l)})^{-\frac{1}{2}}.$$

$\mathbf{o}^{(l)} \to \mathbf{a}^{(l-1)}$:

$$\frac{\partial E}{\partial \mathbf{a}_m^{(l-1)}} = \frac{\partial E}{\partial \mathbf{o}_m^{(l)}} \mathbf{W}_m^{(l)T} + (\frac{\partial E}{\partial \mathbf{o}_s^{(l)}} \mathbf{W}_s^{(l)T}) \circ 2\mathbf{a}_m^{(l-1)}$$

$$\frac{\partial E}{\partial \mathbf{a}_s^{(l-1)}} = \frac{\partial E}{\partial \mathbf{o}_s^{(l)}} \mathbf{W}_s^{(l)T} + \frac{\partial E}{\partial \mathbf{o}_s^{(l)}} (\mathbf{W}_m^{(l)} \circ \mathbf{W}_m^{(l)})^T.$$

$\mathbf{a}^{(l)} \to \mathbf{o}_c^{(l)}$:

$$\frac{\partial E}{\partial \mathbf{o}_c^{(l)}} = -r(\exp(-\tau) - 1)\frac{\partial E}{\partial \mathbf{a}_m^{(l)}} \circ \exp((\exp(-\tau) - 1)\mathbf{o}_c^{(l)})$$

$$+ r^2 \frac{\partial E}{\partial \mathbf{a}_s^{(l)}} \circ ((\exp(-2\tau) - 1)\exp((\exp(-2\tau) - 1)\mathbf{o}_c^{(l)})$$

$$- 2(\exp(-\tau) - 1)\exp(2(\exp(-\tau) - 1)\mathbf{o}_c^{(l)}))$$

$\mathbf{o}^{(l)} \to \mathbf{W}^{(l)}, \mathbf{o}^{(l)} \to \mathbf{b}^{(l)}$:

The gradients with respect to the mean-variance pairs:

$$\frac{\partial E}{\partial \mathbf{W}_m^{(l)}} = \mathbf{a}_m^{(l-1)T} \frac{\partial E}{\partial \mathbf{o}_m^{(l)}} + (\mathbf{a}_s^{(l-1)T} \frac{\partial E}{\partial \mathbf{o}_s^{(l)}}) \circ 2\mathbf{W}_m^{(l)}$$

$$\frac{\partial E}{\partial \mathbf{W}_s^{(l)}} = \mathbf{a}_s^{(l-1)T} \frac{\partial E}{\partial \mathbf{o}_s^{(l)}} + (\mathbf{a}_m^{(l-1)} \circ \mathbf{a}_m^{(l-1)})^T \frac{\partial E}{\partial \mathbf{o}_s^{(l)}}$$

$$\frac{\partial E}{\partial \mathbf{b}_m^{(l)}} = \frac{\partial E}{\partial \mathbf{o}_m^{(l)}}$$

$$\frac{\partial E}{\partial \mathbf{b}_s^{(l)}} = \frac{\partial E}{\partial \mathbf{o}_s^{(l)}}$$

The gradients with respect to the proxy natural parameters:

$$\frac{\partial E}{\partial \mathbf{W}_c^{(l)}} = \frac{\partial E}{\partial \mathbf{W}_m^{(l)}} \circ \frac{1}{\mathbf{W}_d^{(l)}} + \frac{\partial E}{\partial \mathbf{W}_s^{(l)}} \circ \frac{1}{\mathbf{W}_d^{(l)2}}$$

$$\frac{\partial E}{\partial \mathbf{W}_d^{(l)}} = -\frac{\partial E}{\partial \mathbf{W}_m^{(l)}} \circ \frac{\mathbf{W}_c^{(l)}}{\mathbf{W}_d^{(l)2}} - 2\frac{\partial E}{\partial \mathbf{W}_s^{(l)}} \circ \frac{\mathbf{W}_c^{(l)}}{\mathbf{W}_d^{(l)3}}$$

$$\frac{\partial E}{\partial \mathbf{b}_c^{(l)}} = \frac{\partial E}{\partial \mathbf{b}_m^{(l)}} \circ \frac{1}{\mathbf{b}_d^{(l)}} + \frac{\partial E}{\partial \mathbf{b}_s^{(l)}} \circ \frac{1}{\mathbf{b}_d^{(l)2}}$$

$$\frac{\partial E}{\partial \mathbf{b}_d^{(l)}} = -\frac{\partial E}{\partial \mathbf{b}_m^{(l)}} \circ \frac{\mathbf{b}_c^{(l)}}{\mathbf{b}_d^{(l)2}} - 2\frac{\partial E}{\partial \mathbf{b}_s^{(l)}} \circ \frac{\mathbf{b}_c^{(l)}}{\mathbf{b}_d^{(l)3}}$$

## 8 Figures in the Paper

In this section we provide larger versions of figures in the paper for readers' convenience.

Figure 2: Predictive distributions for PBP, BDK, dropout NN, and NPN. The shaded regions correspond to $\pm 3$ standard deviations. The black curve is the data-generating function and blue curves show the mean of the predictive distributions. Red stars are the training data.

Figure 3: Classification accuracy for different variance (uncertainty). Note that '1' in the x-axis means $\mathbf{a}_s^{(L)} \mathbf{1}^T \in [0, 0.04)$, '2' means $\mathbf{a}_s^{(L)} \mathbf{1}^T \in [0.04, 0.08)$, etc.

Figure 4: Reconstruction error and estimated uncertainty for each data point in *Citeulike-a*.

## Footnotes

[1]The relationships between Poisson distributions and Gaussian distributions are described in Theorem 3. The theorem, however, cannot be directly used here since $\mathbf{o}_m^{(l)}$ and $\mathbf{o}_s^{(l)}$ are not identical. This is why we have to resort to the KL divergence.

[2]Note that for Theorem 3 to be valid, $c$ has to be sufficiently large, which is why we do not normalize the word counts as preprocessing and why we use a large $r$ for the activation $v(x) = r(1 - \exp(-\tau x))$.