[Reviews · NeurIPS 2016]

Reviewer 1

Summary

This paper presents natural parameter networks (NPN), a flexible new approach for Bayesian Neural Networks where inference does not require sampling. Distributions over weights and neurons are expressed as members of the exponential family, whose natural parameters are learned. For forward propagation, they are able to compute the integral for calculating the outputs of non-linear activations. With forward propagation made tractable NPNs are trained with an algorithm very similar to backpropagation. They also describe three variations of NPNs with different distributions in the exponential family: Gamma, Gaussian, and Poisson distributions. They evaluate NPNs on MNIST classification, where the NPNs outperform strong baselines. Additionally, the NPNs also show better performance on varying data sizes, indicating robustness to overfitting. However, the authors do not mention if the vanilla dropout NN has the same number of parameters as the NPN. Since the NPN has double the number of parameters as a vanilla NN, they should show the performance of a comparably sized netowrk for the baseline. Experiments also show that NPNs are better are representation learning where they use the distributions of representations learned for a link prediction task. Again, it is not clear if the sizes of the models are comparable.

Qualitative Assessment

I like this paper. The approach is interesting and shows promisze. I would like the authors to shed light if they are comparing networks with the same sizes. If not, then they should repeat the experiments with equally sized networks.

Confidence in this Review

2-Confident (read it all; understood it all reasonably well)


Reviewer 2

Summary

The paper presents Natural Parameter Networks, a class of Bayesian neural networks which yields a predictive distribution by forward propagating distributions in the exponential family via a sequence of linear (whose parameters are also variables in the exponential family) and (element wise) non-linear transformations. One of the main contributions of the paper is in deriving novel non-linear activation functions for which the first two sufficient statistics can be computed in closed form for input distributions in e.g. Gaussian, Gamma or Poisson distributions. Experiments consists of a toy regression task, MNIST classification as a function of the number of labeled training examples and a link rank prediction task which can exploit variance of the intermediate representation obtained by NPN auto-encoders.

Qualitative Assessment

The paper presents a novel and potentially impactful way of learning uncertainty over model parameters. The derivation of novel activation functions for which first and second moments are computable in closed forms (for distributions in the exponential family) appears to be the main (novel) contribution, as this is what allows forward propagation of exponential distributions in the network, and learning of their parameters via backprop. The work does bear some resemblance to earlier work on “Implicit Variance Networks” Bayer et al. which ought to be discussed. On a technical level, the method appears to be effective and the authors empirically verify that: (1) the method is robust to overfitting (2) predictive uncertainty is well calibrated and (3) that propagating distributions over latent states can outperform deterministic methods (e.g. representations extracted by auto-encoders). The fact that these second order representations outperform those of VAE is somewhat more surprising and may warrants further experimentation: this would imply that the approximation used by the VAE at inference, is worse than the approximation made by NPN that each layer’s activation belongs to the exponential family. I do not agree with the authors however that approaches based on variational inference are limited to Gaussian distributions (line 40): the reparametrization trick used by BBB extends to any distribution having location-scale parameters, e.g. logistic, Laplace, student-t, etc. My main concern with the paper is is the lack of shared baselines, between PBP, BBB and BDK. NPN being submitted a year after the publication of these algorithms, it is regrettable that the authors did not make a more conscious effort to evaluate NPN on the same experiments found in these papers. In particular, the bandit task employed by BBB or the Boston Housing dataset would have been appropriate. As is, we can only draw limited comparisons between NPN and competing methods which may limit the impact of the paper. The paper would also benefit from an in-depth analysis of the method. In particular, I would like to see how the main approximation made by NPN (which approximates each layer as an exponential distribution) scales with depth. I would expect this approximation to become worse with depth, as the computation becomes increasing linear. The paper is relatively clear and well written, though I would recommend perhaps keeping only one of {3.1, 3.2, 3.3} as an illustration, and moving the rest to supplemental. This would allow moving some of the experimental results currently in Section 4 of the Supplemental into the main text. Notation could also be generally tightened. Details: * in Theorem 1, is the activation computed “element-wise” over sufficient statistics (e.g. “v_i(x) = “) or is there a missing summation over index i in the argument to exp() ?

Confidence in this Review

2-Confident (read it all; understood it all reasonably well)


Reviewer 3

Summary

An implementation of a Bayesian deep neural network is outlined, where distributions over the weights, outputs and activations are endowed with exponential family distributions. Starting from the inputs, the uncertainty is propagated in a feed-forward manner. Due to nonlinear activations, one has to represent resulting non-exponential family density with moment-matching, i.e. the propagation is approximate in a way that is reminiscent of Expectation Propagation. The usefulness of the second-order information provided by the exponential family representation is demonstrated in cases where we don't have a lot of data or in unsupervised representation learning via an autoencoder.

Qualitative Assessment

I like the overall direction this paper is going in. It is very important to be able to model uncertainty in deep neural networks. However, the approach given in this paper is somewhat incomplete. It doesn't follow the usual Bayesian setup of placing a prior on weights and computing/approximating a posterior given the data. Instead it tries to directly optimize the individual exponential family parameters, sidestepping the central Bayesian update paradigm. I would normally be somewhat ok with this if convincing experimental results were provided. But the results section is quite weak, only 2 superficial experiments are done and the gains seem modest at best. If one could compute a good posterior distribution over weights, then the propagation methods used here would be quite useful however. So in this sense, this paper is useful.

Confidence in this Review

2-Confident (read it all; understood it all reasonably well)


Reviewer 4

Summary

The paper presents a new Bayesian treatment for probabilistic neural networks. The approach allows to specify and estimate distributions within the exponential family for all parameters of the model, as well as its inputs an outputs. Besides, the framework allows for multi-layer architectures and nonlinear activation functions, akin to standard neural networks. With their approach, the authors aim to alleviate two well-know problems of standard neural networks, namely, overfitting when data is insufficient and lack of flexibility in terms of available distributions for weights and neurons. The authors formulate neural networks in terms of parameters' distributions rather than the parameters themselves, which in principle allow the model to capture a richer representation of the data by modeling uncertainty. For additional flexibility, nonlinear activation functions are defined for a variety of distributions within the exponential family. Learning is done via back-propagation and the authors provide details for 3 instances of their framework, namely, gamma, Gaussian and Poisson. Experiments focus on handwritten digits classification (MNIST) and second-order representation learning for text documents using bag-of-words representations.

Qualitative Assessment

My main concern with the paper, on a conceptual level, is that the authors claim to use a Bayesian treatment of neural networks however, the authors make no mention whatsoever to posterior/predictive distributions and the only mention to priors is reserved to Gaussian distributions in the context of regularization. Also, all parameters of the model (weights and neurons) are specified via fully factorized univariate distributions. How is this different from a straightforward mean-filed variational Bayes treatment of neural networks with moment matching for activation functions? One of the most appealing elements of Bayesian analysis is the ability of estimating posterior/predictive distributions to quantify uncertainty via capturing the covariance structure of the data, however such task is very difficult with fully factorized specifications as it is well-known from the variational inference literature. In the experiments, it is not clear to me what is the added value of the approach presented by the authors for three reasons: i) it is difficult to evaluate the significance of the results in Table 3 without error bars. Besides, for Size = 100, although NPN outperforms Dropout, it is not entirely obvious that the latter is overfitting considerably more than the former. ii) Why are the results for unsupervised models evaluated in terms on (indirect measure) link prediction rather than (direct measure) say perplexities or marginal likelihood estimates? iii) One of the main points of the authors is that standard neural networks are limited in terms of distribution choices, yet the authors show that on MNIST a Gaussian model outperforms the Gamma model which in principle is better suited for non-negative data. Besides on the unsupervised task, the Gaussian model compares favorably against the Poisson model, but again results in Table 4 are difficult to interpret as they indirectly quantify the representational capabilities of the models being considered. It would be nice to get some clarity in the following points: - How are the parameters of the activation functions in Table 3 selected/tuned in practice. - How good is the approximation in (5), from the supplementary material, it seems that (5) comes from a well-known result for the probit link function, however, no details are provided about the accuracy of the approximation. - In (6), what prevents the variance, a_s, from becoming a negative number? and if that is the case how it is handled in practice? - In the Gaussian NPN model, outputs, y, are assumed Gaussian with variance epsilon, how is epsilon selected and how it affects performance? - For Poisson NPN, how are counts obtained from the (continuous) approximation proposed by the authors? - The authors should avoid using the term calibration so liberally, as the authors do not provide results to directly ascertain calibration, instead results simply indicate correlation between uncertainty (variance) and accuracy.

Confidence in this Review

2-Confident (read it all; understood it all reasonably well)


Reviewer 5

Summary

This paper proposed a class of probabilistic neural networks called NPN, which regards the weights and neurons as arbitrary exponential-family distributions.

Qualitative Assessment

The method in this paper is a commen application of BNN. It discussed gamma, gaussian and poisson distributions but it didn't compare them in experiments.

Confidence in this Review

2-Confident (read it all; understood it all reasonably well)